# Physical association between a novel plasma-membrane structure and centrosome orients cell division

Takefumi Negishi[1,2]*, Naoyuki Miyazaki[3], Kazuyoshi Murata[3], Hitoyoshi Yasuo[2]*, Naoto Ueno[1,4]*

[1]Division of Morphogenesis, National Institute for Basic Biology, National Institutes of Natural Sciences, Okazaki, Japan; [2]Laboratoire de Biologie du Développement de Villefranche-sur-mer UMR7009, Observatoire Océanologique, Sorbonne Universités, UPMC Université Paris 06, CNRS, Villefranche-sur-Mer, France; [3]National Institute for Physiological Sciences, National Institutes of Natural Sciences, Okazaki, Japan; [4]Department of Basic Biology, School of Life Science, The Graduate University for Advanced Studies, Okazaki, Japan

**Abstract** In the last mitotic division of the epidermal lineage in the ascidian embryo, the cells divide stereotypically along the anterior-posterior axis. During interphase, we found that a unique membrane structure invaginates from the posterior to the centre of the cell, in a microtubule-dependent manner. The invagination projects toward centrioles on the apical side of the nucleus and associates with one of them. Further, a cilium forms on the posterior side of the cell and its basal body remains associated with the invagination. A laser ablation experiment suggests that the invagination is under tensile force and promotes the posterior positioning of the centrosome. Finally, we showed that the orientation of the invaginations is coupled with the polarized dynamics of centrosome movements and the orientation of cell division. Based on these findings, we propose a model whereby this novel membrane structure orchestrates centrosome positioning and thus the orientation of cell division axis.

*For correspondence:
negishi0525@gmail.com (TN);
yasuo@obs-vlfr.fr (HY); nueno@
nibb.ac.jp (NU)

**Competing interests:** The authors declare that no competing interests exist.

## Introduction

The orientation of the mitotic division axis has been proposed to control tissue morphogenesis as well as cell fate determination, thus playing an important role in shaping embryonic forms (*Lu and Johnston, 2013*; *Moorhouse and Burgess, 2014*; *Siller and Doe, 2009*). The mechanism determining the orientation of the mitotic spindle has been extensively studied in both cultured and embryonic cells and precise molecular processes are well understood (*Bell et al., 2015*; *Cao et al., 2010*; *Delaval et al., 2011*; *Kiyomitsu and Cheeseman, 2012*; *Woolner and Papalopulu, 2012*; *Zheng et al., 2010*). One strategy to achieve this is the control of centrosome dynamics. Centrosome works as a microtubule-organizing center (MTOC) in animal cells and consists of a pair of mother and daughter centrioles, which are distinct in both structure and age (*Azimzadeh and Bornens, 2007*). Following duplication and migration, the two centrosomes become aligned to serve as spindle poles during mitosis. Thus, the axis of centrosome alignment is frequently consistent with mitotic spindle orientation unless additional constraints such as cell shape exist to alter the spindle orientation (*Gibson et al., 2011*; *Minc et al., 2011*). Specific mother-daughter centrosome inheritance coupled with asymmetric cell division is a highly conserved phenomenon (*Yamashita, 2009*). It has been first reported in budding yeast that the 'old' spindle pole body corresponding to the mother centrosome, segregates into the bud (*Pereira et al., 2001*). *Drosophila melanogaster* male

**eLife digest** An animal develops from a single fertilized egg cell. Several rounds of cell division then occur to create new cells and form an embryo. Often, the direction of cell division is oriented, rather than random. In other words, the positioning of the two new daughter cells is highly organized during cell division. This orientation of the direction of cell division is also important for shaping the body's tissues.

Animal cells contain a structure called the centrosome that helps to regulate cell division (amongst other roles). Just before a cell divides, the centrosome duplicates itself and the copies move toward opposite ends of the cell. A structure called the mitotic spindle then forms out of the centrosomes and ensures that the newly forming cells contain the correct amount of genetic material. The orientation of the spindle specifies where the cell splits into two, and this orientation is ultimately governed by the position of the centrosomes inside the cell. However, it is not fully understood how cells position their centrosomes.

Sea squirts are simple marine animals that are well suited for studying cell division, in part because their embryos consist of a small number of cells. Negishi et al. have now studied the final cell division cycle of the outer cells of sea squirt embryos, during which nearly all the cells divide in the same direction – along an axis that stretches from the embryo's head to its tail. This revealed that before a spindle forms in these cells, the cell membrane at the rear end of each cell is pulled into the cell, forming an "invagination" that elongates along the head-to-tail axis.

The finger-like membrane invagination captures the centrosome and pulls it towards the rear end of the cell. Following this, the centrosome duplicates and the new centrosomes move until they are aligned with the membrane invagination. Once both centrosomes are aligned correctly, the spindle forms.

Thus, membrane invaginations serve to position centrosomes. The next steps are to identify the molecules that allow membrane invaginations and centrosomes to interact with each other and to determine the forces that place centrosomes in their correct location.

germline stem cells and neuroblasts have contributed to our understandings of the molecular mechanisms underlying the asymmetric migration of the duplicated centrosomes during interphase. In male germline cells, membrane localized Adenomatous polyposis coli 2 (Apc2) and the *Drosophila* Par-3 homolog, Bazooka, associated with E-cadherin, tethers one centrosome adjacent to the niche, called the hub, and consequently ensures spindle orientation and asymmetric stem cell division (*Inaba et al., 2015a*; *Yamashita et al., 2003*). In male germline cells, it is the mother centrosome with stable astral microtubules which is anchored near the hub (*Yamashita et al., 2007*). In neuroblasts, the centrosome with the higher MTOC activity remains in the neuroblast following asymmetric cell division (*Rebollo et al., 2007*). In contrast to the male germ line, it is the daughter centrosome that is retained in the stem cell (*Conduit and Raff, 2010*; *Januschke et al., 2011*). Centrobin, associated with the daughter centrosome, was found to be responsible for this oriented cell division (*Januschke et al., 2013*). In both cell systems, the centrosome with a higher MTOC activity is less motile and is inherited by the stem cell (*Pelletier and Yamashita, 2012*).

In addition to a role in spindle orientation, the centrosome also has an important role in cilia formation. During ciliogenesis, the mother centriole converts into the basal body in a quiescent (G0 phase) or interphase (G1 phase) cell to nucleate a primary cilium. Following re-entry or progression of the cell cycle, the primary cilium is disassembled and the basal body/mother centriole is reused for mitotic spindle formation (*Kobayashi and Dynlacht, 2011*). It is unclear how the centrosome transition is coordinated between cilia and spindle.

In this study, we use embryos of ascidian, belonging to the phylum Tunicata, a sister group of the vertebrates (*Satoh et al., 2014*). Ascidian embryos are ideally suited to study mechanisms of cell division because of their invariant cleavage pattern and the small number of cells that form their bodies (*Conklin, 1905*; *Nishida, 1986*). The pattern of cell division is highly conserved among different ascidian species (*Conklin, 1905*; *Lemaire et al., 2008*; *Zalokar and Sardet, 1984*). This implies robust mechanistic constraints on the cell division patterns of ascidian development. Several studies,

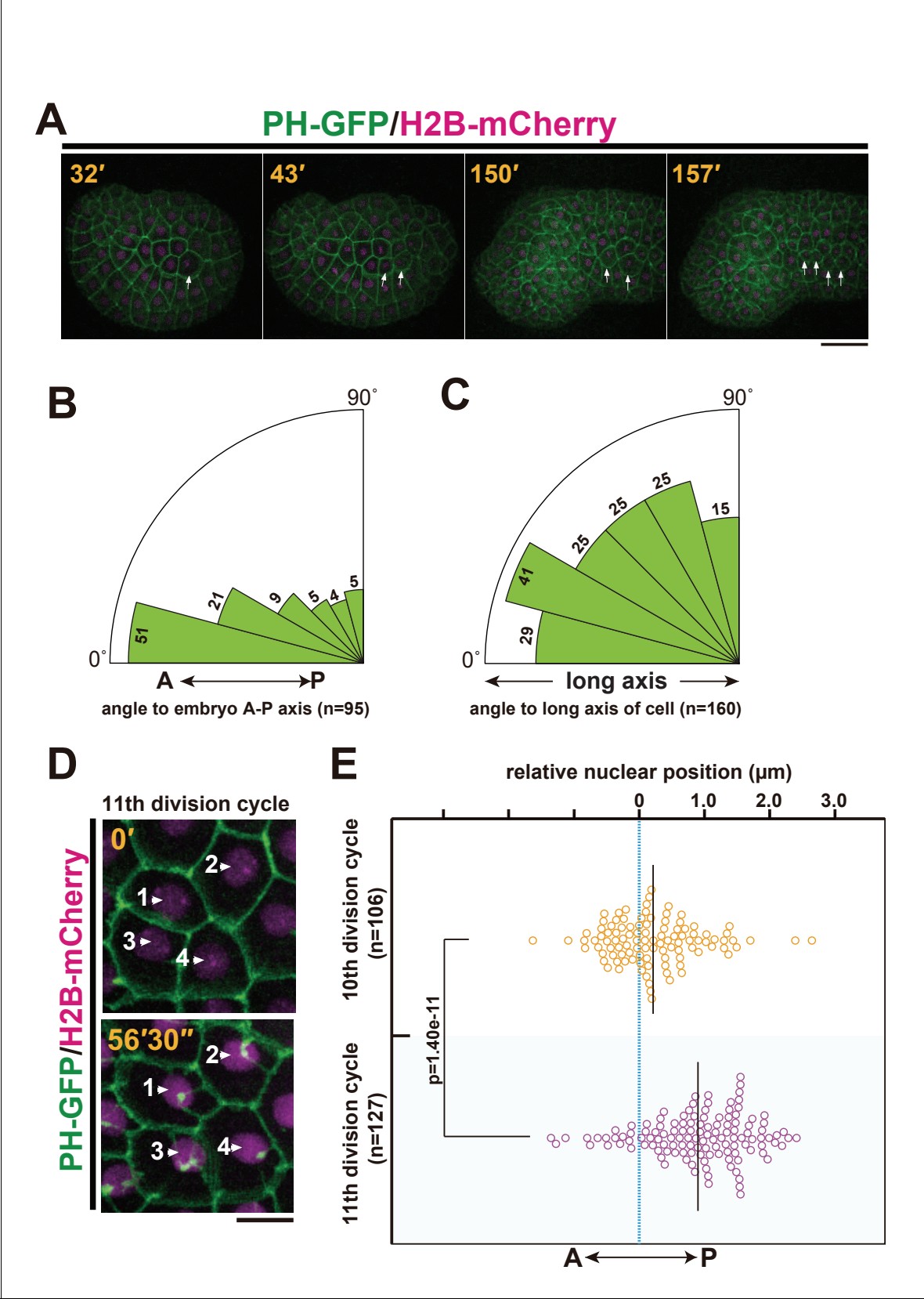

**Figure 1.** *Ciona intestinalis* epidermal cell mitosis and the posterior nuclear positioning prior to the final cell division. (**A**) Representative epidermal cell divisions (white arrows) from the 10th to the 11th cell cycle in an embryo expressing PH-GFP/H2B-mCherry; frames are from *Video 1*. Elapsed time: 32′–
*Figure 1 continued on next page*

*Figure 1 continued*

43' shows the 10th cell division, while 150'–157' shows the 11th cell division. During this process, the embryo changes into a tadpole-shape consisting of a head and tail. Anterior: left. Dorsal: upper. Bar: 30 μm. (B, C) In the final mitotic division, ascidian epidermal cells do not divide following the Sachs's and Herwig's rules. (B) Rose diagram showing the angle of the cell division axis relative to the embryonic A-P axis in the 11th cell division. Following the 10th cell division, we selected daughter cells that were produced via an A-P oriented cell division with less than 30° of the cell division axis relative to the embryonic A-P axis and then measured their cell division angle at the 11th cell division. n = 95; cells from three embryos were used. (C) Rose diagram showing the angle of the cell division axis relative to the long axis of the cell in the last cell division. n = 160 cells from three embryos. (D) Representative frames of a 4D confocal dataset imaging epidermal cells of an embryo expressing PH-GFP/H2B-mCherry. Nuclei show a posteriorly biased positioning prior to M-phase. Numbered arrows indicate the same nucleus in the sequence. Time elapsed from the start of recording is shown in orange. Bar: 10 μm. (E) Bee-swarm plots indicating the nuclear position relative to the centre of the cell along the embryonic A-P axis, measured just before the breakdown of the nuclear membrane, in the 10th and 11th cell-division cycles. n = 106 cells each at the 10th and 11th division cycles from three embryos. Black lines show the average nuclear position relative to the centre of the cell (blue dotted line); the average positions were 0.18 and 1.0 μm toward the posterior side at the 10th and 11th division cycle, respectively. p-values were obtained using the Welch's t-test.

including our own, have reported unique mechanisms of spindle orientation in ascidian embryos (*Kumano et al., 2010*; *Nakamura et al., 2005*; *Negishi and Yasuo, 2015*; *Negishi et al., 2007*; *Nishikata et al., 1999*; *Prodon et al., 2010*). In this study, we focused on embryonic epidermal cells in the cosmopolitan ascidian, *Ciona intestinalis*. It was previously reported that almost all epidermal cells cleave along the anterior-posterior (A–P) axis at the last (11th) division when the *Ciona* embryo starts shaping into a tadpole larval form (*Ogura et al., 2011*). We describe here a novel membrane structure that may control centrosome dynamics including ciliary positioning and spindle orientation during this last cell division of the epidermal cells.

## Results

### A unique membrane structure during interphase of ascidian epidermal cells undergoing oriented cell division

The epidermal cell lineage of ascidians is known to divide alternately along the A-P and medial-lateral (M–L) axes during early cleavage stages (*Nishida, 1994*). The perpendicular shift of the cell division axis during successive rounds of cell division is thought to result from a 90° translocation of the duplicated centrosomes around the nucleus to the opposite directions (Sach's rule) (*Mardin and*

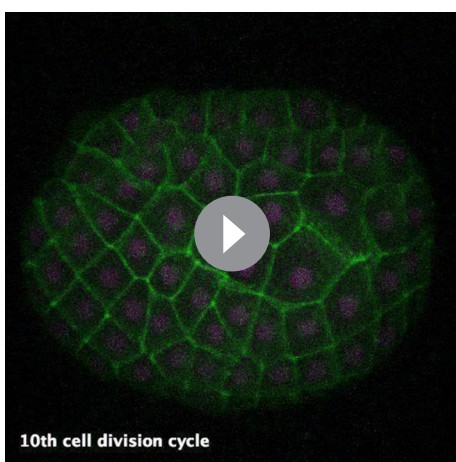

**Video 1.** (10 fps) A low-magnification, low-resolution time-lapse movie of an embryo expressing PH-GFP (green) and H2B-mCherry (magenta), made with the maximum-intensity projection of the confocal microscopy data. Lateral view: facing; anterior: left. Ten minutes are compressed to one second. White arrows: representative cells. This video is related to *Figure 1A*.

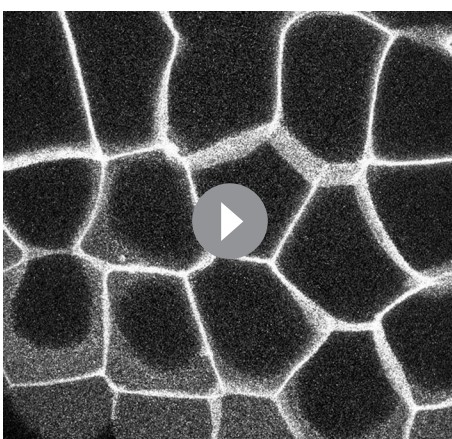

**Video 2.** (7 fps) A high-magnification, high-resolution time-lapse movie of an embryo expressing PH-GFP, made with the maximum-intensity projection of the confocal microscopy data. Four minutes and 40 s are compressed to one second. Anterior: left. This video is related to *Figure 2B*.

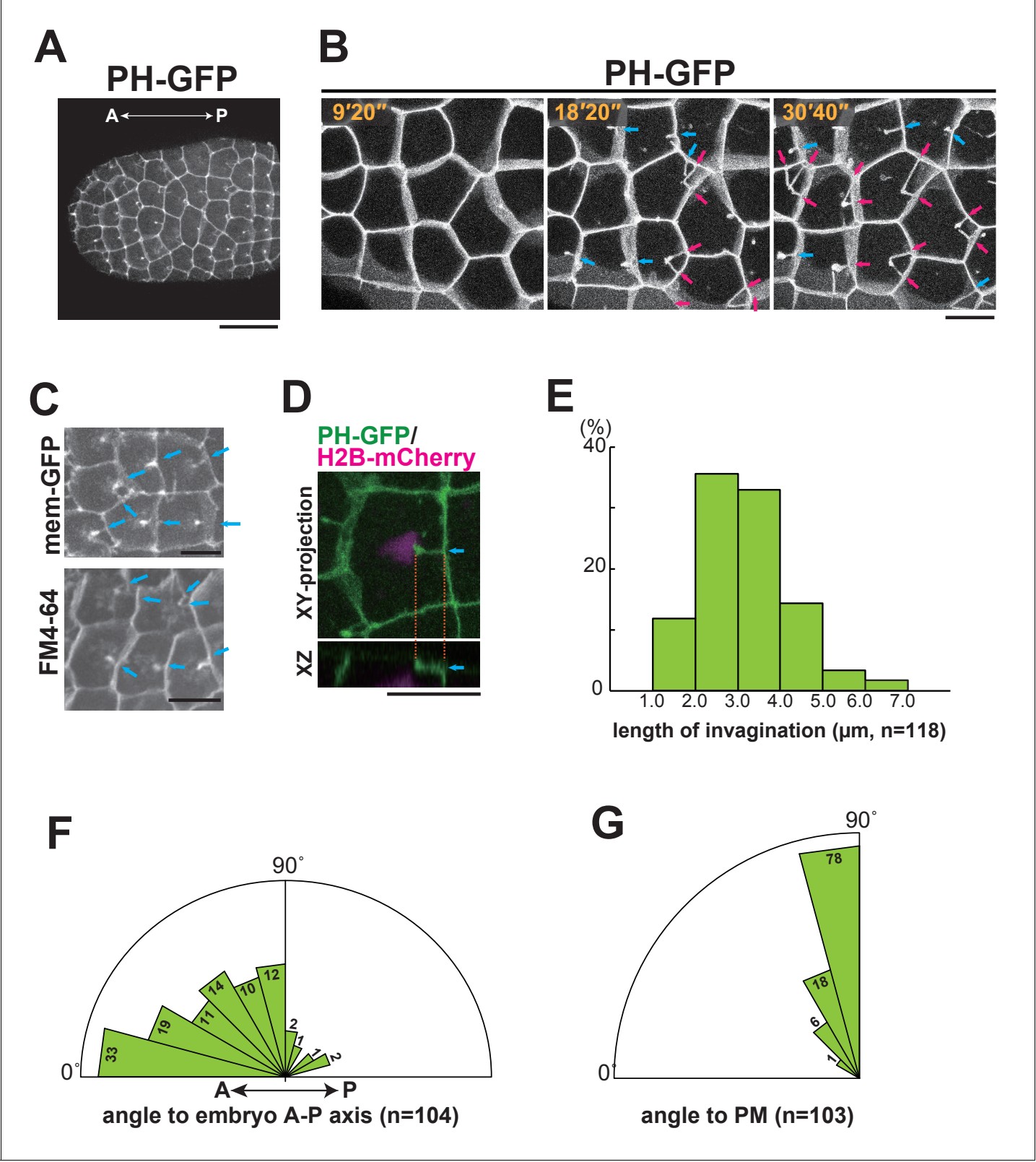

**Figure 2.** Characterisation of the invaginating membrane structure. (**A**) Low-magnification image of epidermal cells in a normal *Ciona intestinalis* embryo expressing PH-GFP, during the 11th cell division cycle. A maximum-intensity projection image of the confocal stack is shown. Anterior: left; ventral side: facing. Black bar: 30 μm. (**B**) High-magnification images of epidermal cells in a normal embryo expressing PH-GFP, during the 11th cell-

*Figure 2 continued on next page*

*Figure 2 continued*

division cycle. Images are from **Video 2**; elapsed times are indicated. Anterior: left. Both blue and red arrows indicate membrane invaginations; red arrows show invaginations forming a wedge shape. (**C**) Membrane invaginations in epidermal cells in a membraneGFP-expressing embryo (upper panel) and a FM4-64 stained embryo (lower panel). Anterior: left; ventral: facing. Bars: 10 µm. Blue arrows: invaginations. (**D**) A membrane invagination formed near the apical cortex in an epidermal cell expressing PH-GFP/H2B-mCherry. XY-projection panel shows a maximum-intensity projection of the confocal stack; the YZ panel was reconstructed from the same confocal data set. Blue arrows: invaginations. Orange dotted lines indicate the same invagination in both panels. Bar: 10 µm. (**E**) Histogram showing the distribution of invagination length, calculated from confocal images. n = 118 invaginations counted from three embryos. (**F**) Rose diagram showing the angle of the invagination relative to the embryonic A–P axis. Almost all of the invaginations had an angle <90°, meaning they formed toward the anterior. n = 104; invaginations counted in three embryos. (**G**) Rose diagram showing the angle of invagination relative to the plasma membrane from which it arises, indicating that the invaginations extends perpendicular to the lateral membrane. n = 103 invaginations counted in three embryos.

*Schiebel, 2012*; *Strome, 1993*). This alternating 90° shift of the division axis tends to follow the long-axis rule based on cell shape (Hertwig's rule) (*Hertwig, 1984*; *Minc et al., 2011*). With live imaging analysis of epidermal cell division, we confirmed that almost all epidermal cells divide along the A–P axis at the last (11th) division as reported previously (*Ogura et al., 2011*). This oriented cell division occurs regardless of the cell shape and whether the 10th cell division occurred along the A–P or M-L axis (*Figure 1A–C*, *Video 1*). Additionally, we found, during the 11th (but not the 10th) cell cycle, that the nucleus of the epidermal cells gradually shifts toward the posterior side (*Figure 1D,E* and *Video 1*).

To analyse the epidermal cell morphology during the 10th and 11th cell-division cycles, we used GFP-conjugated Pleckstrin homology domain of PLC1δ1 (PH-GFP) to visualise the plasma membrane (*Audhya et al., 2005*; *Carroll et al., 2003*; *Hurley and Meyer, 2001*). We found a unique structure (membrane invagination) that formed in almost all of the epidermal cells specifically during the interphase of the last (11th) division cycle (*Figure 2A,B*, magenta and blue arrows). Time-lapse observations of cells that had committed to cell division revealed that the invaginating membrane did not adopt a stable, solid form, but rather appeared to be a highly dynamic filamentous structure (*Video 2*). To confirm the structure of the invaginating membranes, we used another membrane binding probe using a domain of c-Ha-Ras, memGFP (*Morita et al., 2012*) or membrane-staining fluorescent dye, FM4-64 (*Figure 2C*), which demonstrated that the invaginations were not an artefact of a specific protein or GFP staining itself. The filamentous structures are located near the apical surface approximately 1.5 µm beneath the apical membrane (base: 1.33 ± 0.56 µm, tip: 0.98 ± 0.37 µm n = 71, *Figure 2D*). The length of the invaginations changed over time, ranging from roughly one-third to half of the cell diameter (2.0–5.0 µm) (*Figure 2E*). Interestingly, some cells that formed a vertex with posterior cells had two or more invaginations (*Figure 2B*, magenta arrows). Quantitative analysis showed that most of the invaginations projected from the posterior or lateral side toward the anterior (*Figure 2F*), at a nearly perpendicular angle to the cell membrane (*Figure 2G*), although transient, randomly projected filamentous membranes were also observed (See *Videos 6* and *8*). These results suggest that the membrane structure forms near the apical cortex and elon-

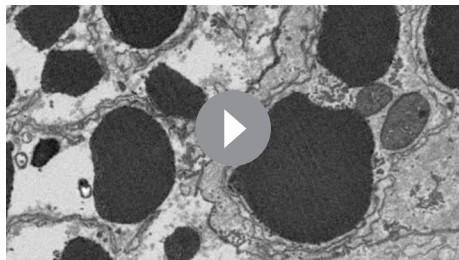

**Video 3.** A video made of serial sections of SBM-SEM images. Blue arrow: invagination. Magenta arrowhead: centrosome. The depth between each frame is 50 nm. This video is related to *Figure 3A*.

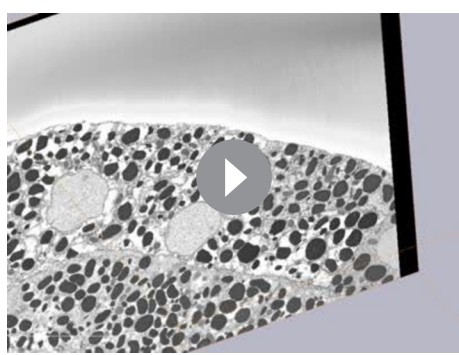

**Video 4.** A movie integrating serial sections of SBF-SEM images and segmentations of the structures.

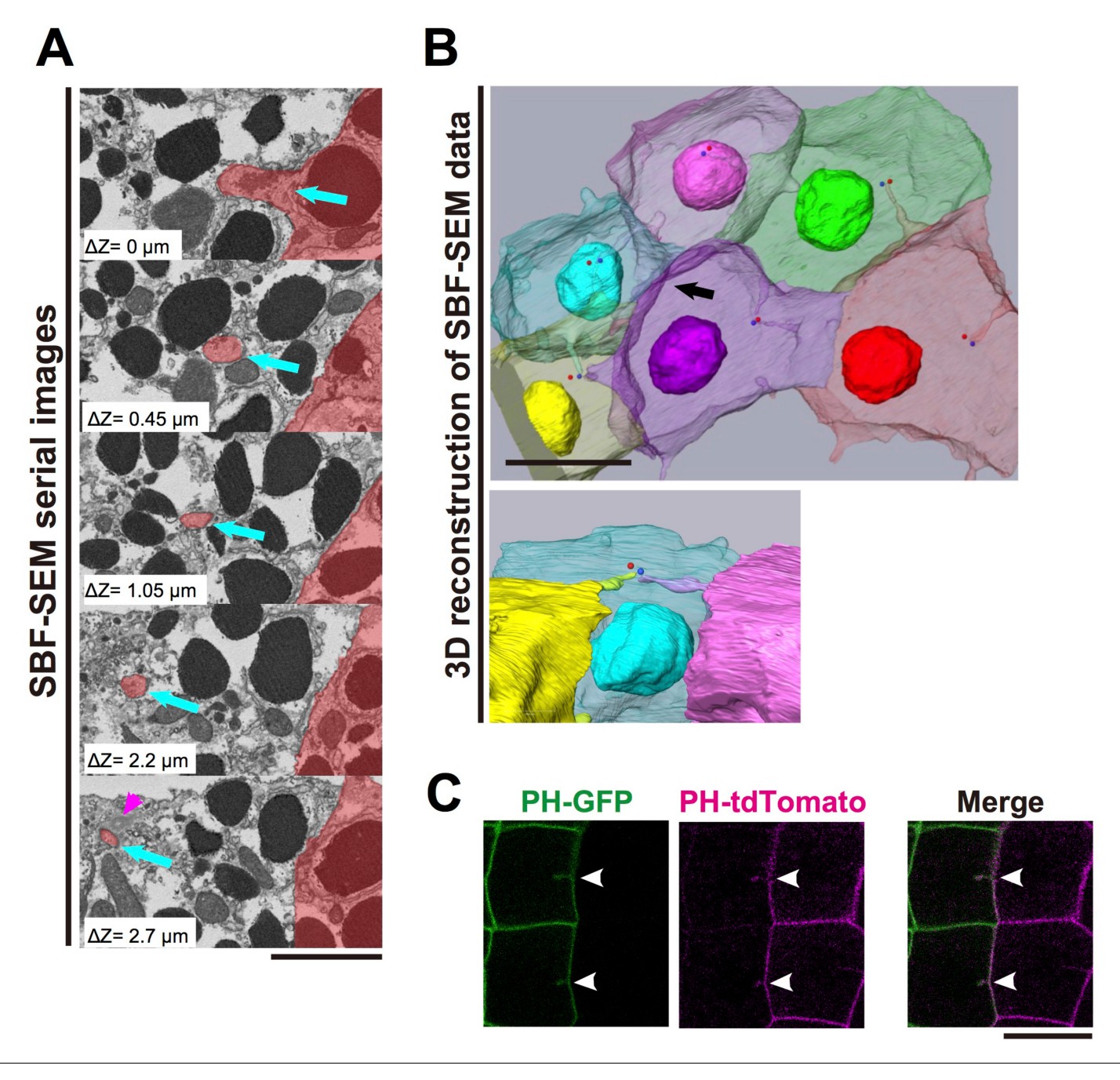

**Figure 3.** The invaginations consist of two plasma membranes and elongate toward the centrosome. (A) Selected z-sections from the SBF-SEM sequence in *Video 3*; ΔZ indicates the depth below the top panel. Blue arrows: invaginations. Magenta arrowhead: centriole. The posterior cell is coloured in red. Bar: 2 μm. (B) A segmentation figure of 3D-reconstructed SBF-SEM data derived from *Video 5*. Lower panel: view from the perspective of the black arrow in the upper panel. Individual cells are labelled in the same colour code in both images. Red and blue balls indicate the position of a pair of centrioles. Bar: 10 μm. (C) Two-colour labelling of the anterior (PH-GFP) and posterior (PH-tdTomato) lineage epidermal cells showing that the invaginations are derived from the plasma membranes of both neighbouring cells. White arrowheads: invaginations on the border that were labelled with both colours. Bar: 10 μm.

gates along the A-P axis. All of the invaginations disappeared just before spindle formation became visible, suggesting that the formation of the invagination is temporally regulated. In fact, the invaginations were observed without exception during the interphase of the 11th cell cycle in epidermal

cells. These results prompted us to investigate whether this novel membrane structure is involved in the spindle orientation.

## Centriole-targeting of the membrane invagination suggests its role in spindle orientation

We next used Serial block-face (SBF) scanning electron microscopy (SEM) (*Denk and Horstmann, 2004*), an advanced 3D electron-microscope technique, to examine the detailed structure of the membrane invaginations. For this analysis, ascidian embryos were fixed at the 11th cell-division cycle. In a series of several hundred sections, we observed fragmented membrane structures directed toward the centrioles (*Figure 3A*, blue arrows, *Video 3*). Notably, 3D reconstructions from serial sections (*Figure 3B*, *Videos 4* and *5*) showed the tip of the invagination approaching the centrioles as if to capture the organelle. These observations raised the interesting possibility that the invagination might physically capture the centrosome and pull it toward the posterior of the cell. SEM images showed that the invaginations were formed of a double-bilayer plasma membrane and that adjacent posterior cell membrane also contributed to this structure (*Figure 3A*, *Videos 3–5*). To confirm this observation, we expressed green (GFP) and red (tdTomato) fluorescence proteins conjugated with the PH domain in cells of the anterior and posterior epidermal lineages, respectively, and observed the membranes of anterior and posterior juxtaposed cells (*Figure 3C*). At the border between two differently labelled cells, the invaginating membranes were labelled with both green and red fluorescence, indicating that the layers of the invaginating membrane were derived from the both of the neighbouring cells (*Figure 3C*, white arrowheads).

We further explored the relationship between the invaginations and centrosomes, which act as a MTOC orchestrating the microtubule dynamics during interphase and mitotic spindle formation. MTOCs in epidermal cells were fluorescently labelled with the microtubule-binding protein ensconsin (E-MAP-115) tagged with tdTomato (*Bulinski et al., 2001*; *Dong et al., 2011*) (*Figure 4A* and *Video 6*). We found that the interphase microtubule array was aligned with the A-P axis with its nucleation site localised at the posterior side, indicating that a stable MTOC was positioned to the posterior of the cells. To record the dynamics of centrosome behaviour in epidermal cells, we decided to use end-binding 3 (EB3), which marks the plus end of the microtubule as it grows away from the organising centre (*Akhmanova and Hoogenraad, 2005*) and highlights the MTOC activity at the centrosome. Before the formation of the invagination at the early interphase of the 11th

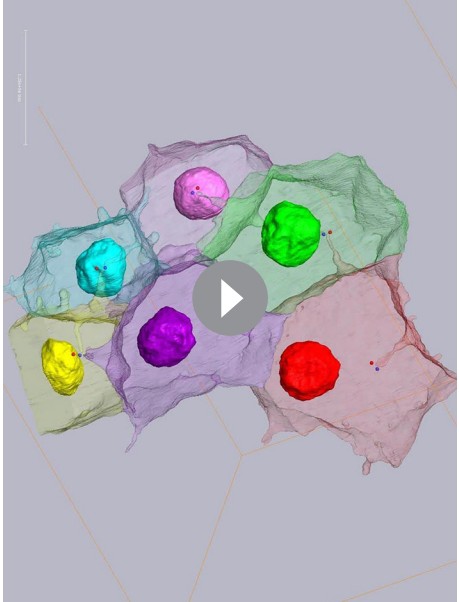

**Video 5.** 3D-segmentations from serial sections of SBM-SEM images: pairs of red and blue balls indicate the positions of centrioles in the cell. This video is related to *Figure 3B*.

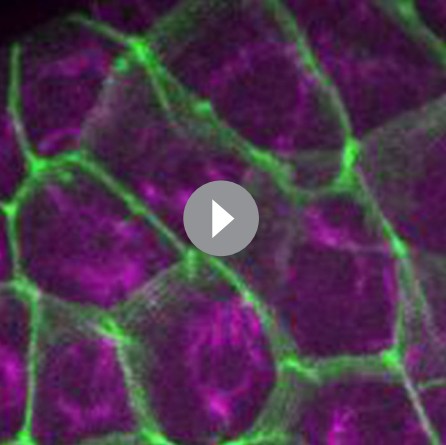

**Video 6.** (10 fps) A time-lapse movie of an embryo expressing PH-GFP (green) and ensconsin-tdTomato (magenta), made with the maximum-intensity projection of the confocal microscopy data. The video starts at the end of the 10th cell division. Five minutes are compressed to one second. Anterior: left. This video is related to *Figure 4A*

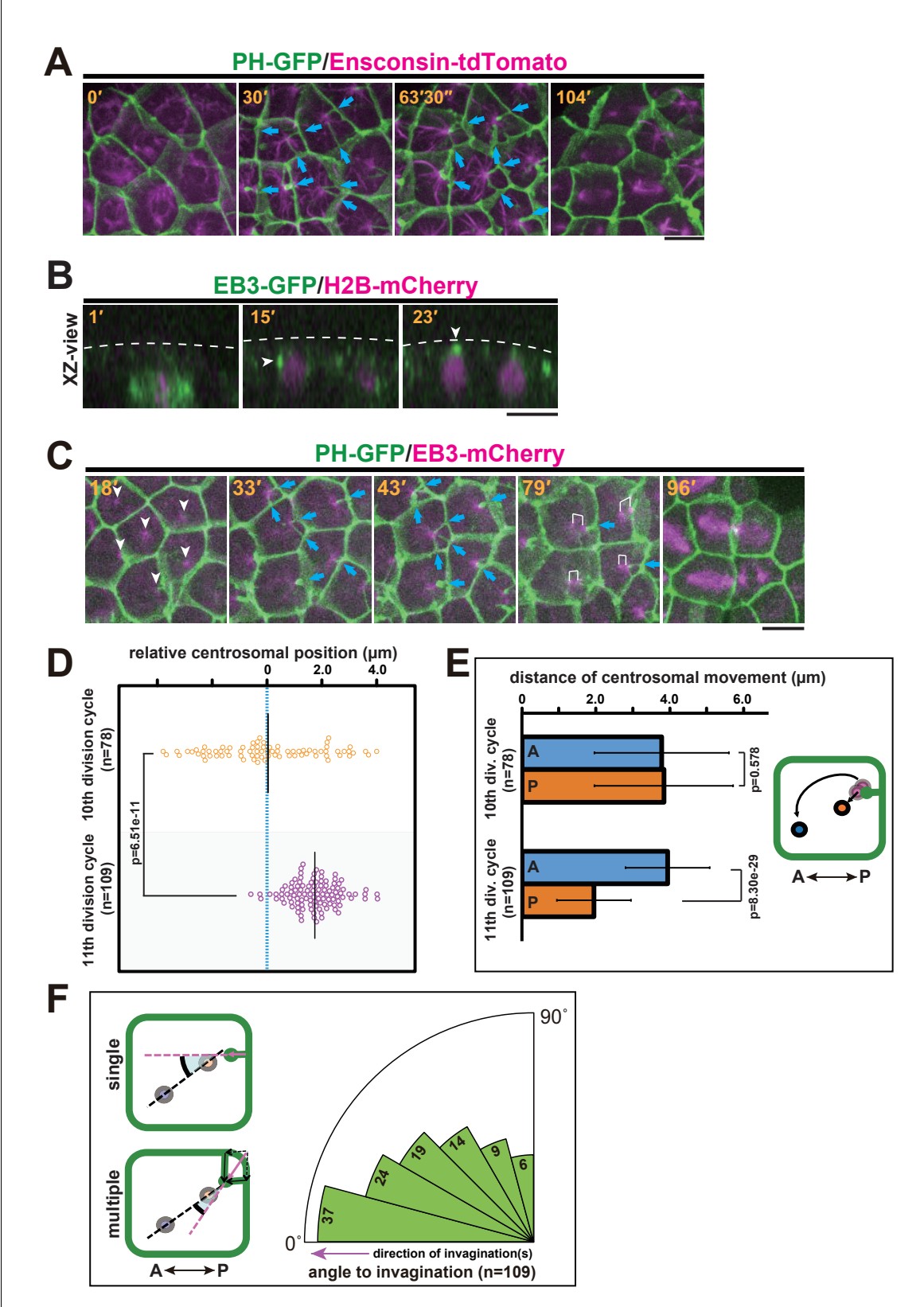

**Figure 4.** Quantitative description of centrosome behaviour in relation to the membrane invaginations. (**A**) Epidermal cells in the 11th cell cycle showing the interphase microtubule array along the A-P axis and membrane invaginations toward the microtubule nucleation site in an embryo

*Figure 4 continued on next page*

*Figure 4 continued*

expressing PH-GFP/Ensconsin-tdTomato. Blue arrows: invaginations. Panels: frames from *Video 6*; elapsed time is indicated. (B, C) Centrosome dynamics in the 11th cell cycle epidermal cells in an embryo expressing EB3-GFP/H2B-mCherry or PH-GFP/EB3-mCherry; elapsed time is indicated in each panel. Bars: 10 μm. (B) XZ-view of epidermal cells in transition from late 10th cell cycle to early 11th cell cycle in an embryo expressing EB3-GFP/H2B-mCherry. Frames were selected from *Video 7*. The focus is on the left daughter cell. Anterior: left. The centrosome (white arrowhead) was located lateral to the nucleus after mitotic division (15′) but moved toward the apical cortex (dotted line). (C) XY-projection images of 11th cell cycle epidermal cells in an embryo expressing PH-GFP/EB3-mCherry, highlighting the elongation of membrane invaginations toward the MTOC. Anterior: left. Each frame was selected from *Video 8*. White arrowheads: microtubule nucleation sites. Blue arrows: invaginations. White brackets: pairs of centrosomes. (D) Centrosome position just before duplication during the 10th (n = 78 cells) and 11th (n = 109 cells) cell cycles. Black bars show the average centrosome position relative to the centre of the cell (blue dotted line); the average centrosome position was 0.03 and 1.7 μm toward the posterior side at the 10th and 11th cell cycles, respectively. p-values were obtained using the Welch's t-test. (E) Migration distance of anterior and posterior centrosomes determined by recording centrosome positions just before duplication and after the migration ceased during the 10th (n = 78 cells from three embryos) and 11th (n = 109 cells from three embryos) cell cycles. Histograms are presented as the mean ± SD, p-values were obtained using the Welch's t-test. Inset: illustration showing how the centrosome behaviours were measured. (F) Relationship between the centrosome axis and the invagination axis. To measure the angle between these axes, we measured the angle of the invagination just before centrosome duplication and the angle of the centrosome axis just after migration (n = 109 cells from three embryos), and calculated the difference between the two angles. For multiple invaginations, a composite vector was used as the angle of the invaginations. Left: illustration showing how two angles were measured. Black dotted line: the centrosome axis. Pink dotted line: the direction of the invagination(s). The bold black arc shows the angle.

cell cycle, EB3-labelled centrosomes were located beneath the apical surface of the epidermal cells (*Figure 4B*, white arrowheads and *Video 7*). We then co-visualised the centrosomes and membrane invaginations by time-lapse imaging of EB3-mCherry/PH-GFP (*Figure 4C* and *Video 8*). These analyses showed that the invaginations appeared to extend toward highly concentrated EB3-mCherry signals over time (*Figure 4C*, 33′–79′). Importantly, the invagination and the centrosome remain associated during posterior displacement. Although during the 10th division cycle, centrosomes were distributed equally along the A-P axis of epidermal cells (*Figure 4D*), in the 11th division cycle following the emergence of the membrane invagination, the distribution of centrosome positioning was strongly biased to the posterior of the cell (*Figure 4C,D*). By the time the invagination disappeared, the centrosome had duplicated; the two centrosomes then migrated asymmetrically until they were aligned with the axis of invagination (*Figure 4C*, white brackets in 79′). Indeed, our quantitative analysis indicated that the posterior-most centrosome migrated a shorter distance than the anterior one in the 11th division cycle, while this is not the case in the previous cell cycle (*Figure 4E*). After the centrosome migration, we measured the angle between the axis of aligned centrosomes and the direction of invagination(s).

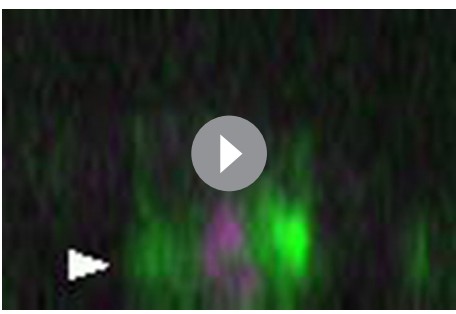

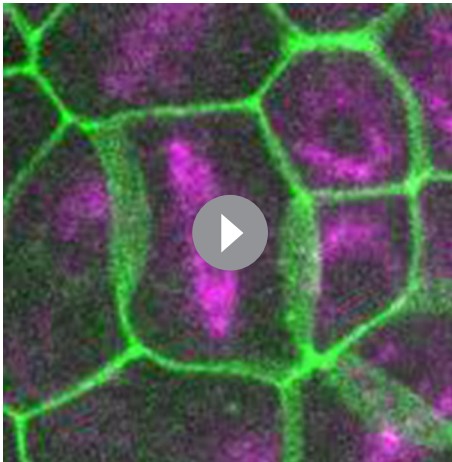

**Video 7.** (7 fps) A time-lapse movie of an embryo expressing EB3-GFP (green) and H2B-mCherry (magenta), made with the reconstructed cross-sections of confocal microscopy data. The video starts at the M phase of the 10th cell division. Four minutes and 40 s are compressed to one second. Apical side: up. This video is related to *Figure 4B*.

**Video 8.** (10 fps) A time-lapse movie of an embryo expressing PH-GFP (green) and EB3-mCherry (magenta), made with the maximum-intensity projection of the confocal microscopy data. The video starts at the end of the 10th cell division. Five minutes are compressed to one second. Anterior is left. This video is related to *Figure 4C*.

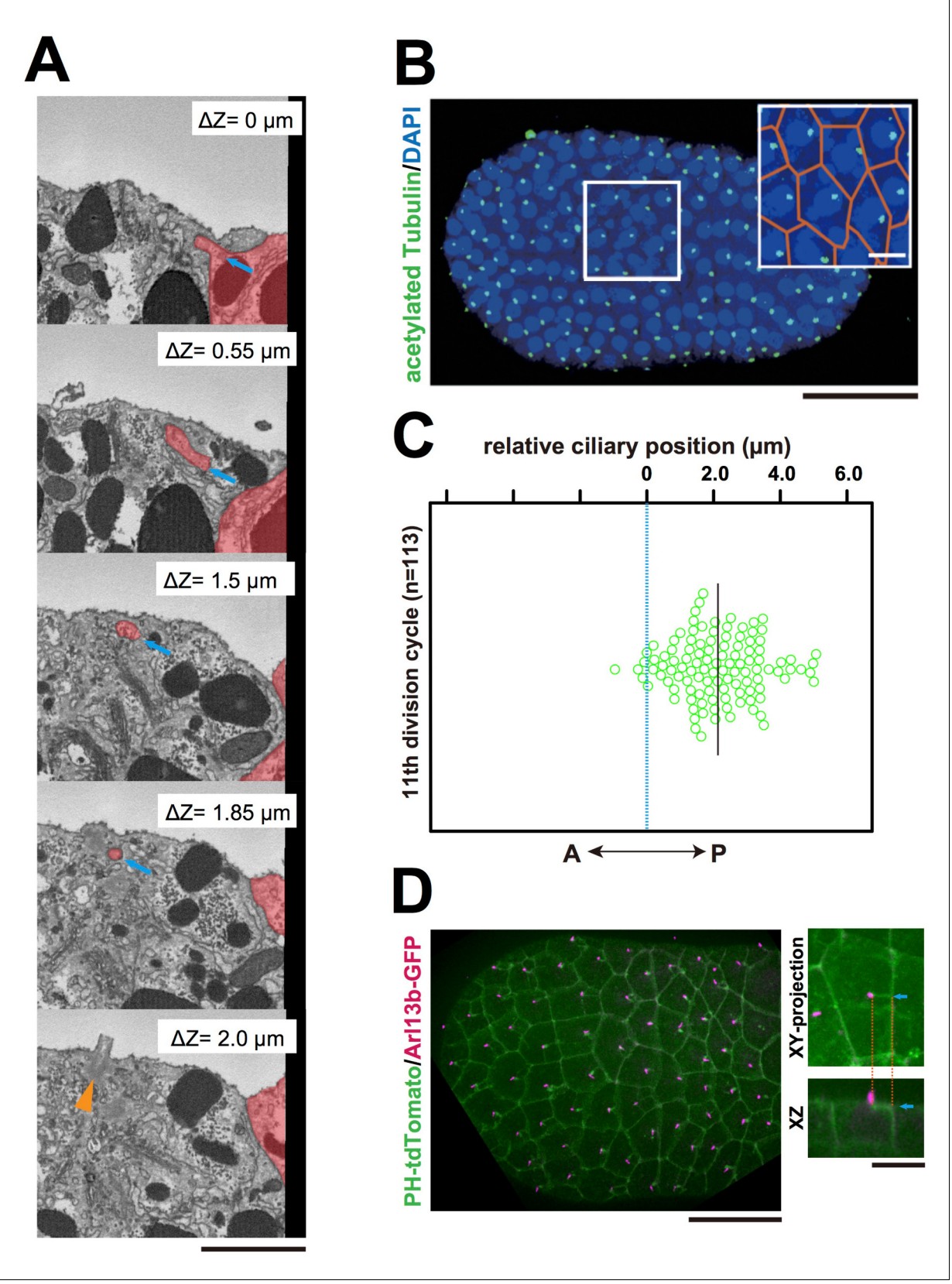

**Figure 5.** The centriole associated with the membrane invaginations carries the primary cilium. (**A**) Selected z-sections from the SBF-SEM sequence in *Video 9*; ΔZ indicates the depth starting from the z-level of the top panel. Blue arrow: invagination. Orange arrowhead: the centriole and cilium

*Figure 5 continued on next page*

*Figure 5 continued*

complex. The posterior cell is tinted in red. Bar: 1 µm. (**B**) Acetylated tubulin immunofluorescence counterstained with DAPI, showing that the epidermal cells of the ascidian embryo contained a primary cilium at the 11th cell cycle stage. A maximum-intensity projection of the confocal z-stack is shown. Black bar: 50 µm. Area in the white square is enlarged in the inset; cell contours were manually outlined in orange. White bar: 10 µm. (**C**) Analysis of the position of cilia in the ascidian epidermal cells during the last cell cycle, showing a tendency to localise to the posterior side, consistent with previous observations (*Thompson et al., 2012*). We measured 113 cilia from three embryos. The black bar shows that, relative to the centre of the cell (blue dotted line), the average cilium position was 2.1 µm toward the posterior side of the cell. (**D**) Membrane invaginations and cilia were observed simultaneously in an embryo expressing PH-tdTomato and ADP-ribosylation factor-like 13b (Arl13b)-GFP; Arl13b labels primary cilia (*Duldulao et al., 2009*; *Paridaen et al., 2013*). The XY-projection panel shows a representative epidermal cell from another embryo. The XZ panel was reconstructed from the same z-stack data used for the XY-projection panel. Blue arrows: invaginations. Orange dotted lines indicate the same invagination in both panels. Bar: 10 µm.

When there were multiple invaginations, we calculated the composition of these structures as a vector. The analysis revealed that the migrated centrosomes were well-aligned along the axis of membrane invagination (*Figure 4F*). Mitotic spindles then form with the two spindle poles aligned along the axis of the invagination (*Figure 4A*, 104'; *Figure 4C*, 79' and 96'). Taken together, these observations suggest that the formation of the invagination may be correlated with the apical and posterior positioning of the centrosomes and the subsequent asymmetric migratory behaviour of the duplicated centrosomes.

Interestingly, serial SBF-SEM sections revealed that primary cilia were located near the centrioles associated with the invaginations (*Figure 5A* and *Video 9*). We also confirmed the posterior localization of cilia in 11th cell cycle epidermal cells in embryos immunostained with anti-acetylated tubulin (*Figure 5B,C*). Posterior cilium positioning was highly reminiscent of the posterior centrosome positioning (*Figure 5D*). We speculate that the primary cilia associated with the membrane invaginations in this study are the same cilia reported previously (*Thompson et al., 2012*). Live imaging of a GFP-tagged ciliary protein, ADP-ribosylation factor-like (ARL) family of small GTPases, Arl13b (*Duldulao et al., 2009*; *Paridaen et al., 2013*), showed that almost all cilia are associated with invaginations (97.8%, 221/226 cells in three embryos, *Figure 5D*). All Arl13b-GFP labelled structures disappeared in the mitotic phase (n = 165 cells). Measuring the distance between the tip of invagination and two centrioles (basal body and daughter) in 3D images reconstructed from SBF-SEM sections revealed that the invagination tip comes in closer proximity to the basal body derived from the mother centriole (*Kobayashi and Dynlacht, 2011*) than to the daughter centriole (9/10 invaginations) (*Figure 6A,B*). These results suggest that the basal body as well as the mother centriole are targets of the invagination.

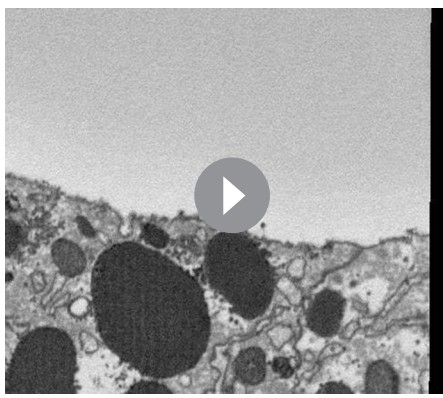

**Video 9.** A video made with serial sections of SBM-SEM images. Blue arrow: the invagination. Orange arrowhead: the cilia and centriole. The depth between each frame is 50 nm. This video is related to *Figure 5A*.

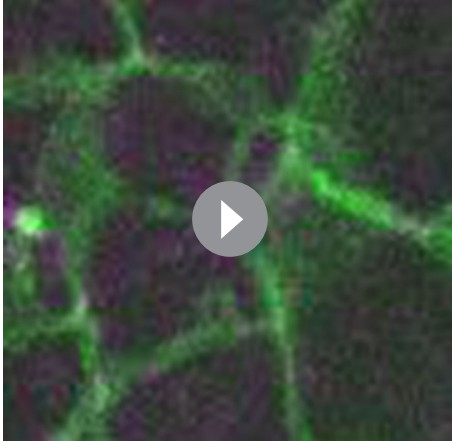

**Video 10.** (12 fps) In a cytochalasin-treated embryo expressing PH-GFP/EB3-mCherry, membranes invaginated from the anterior side and the number of invaginations increased. Anterior: left. Five minutes are compressed to one second. This video is related to *Figure 7A*.

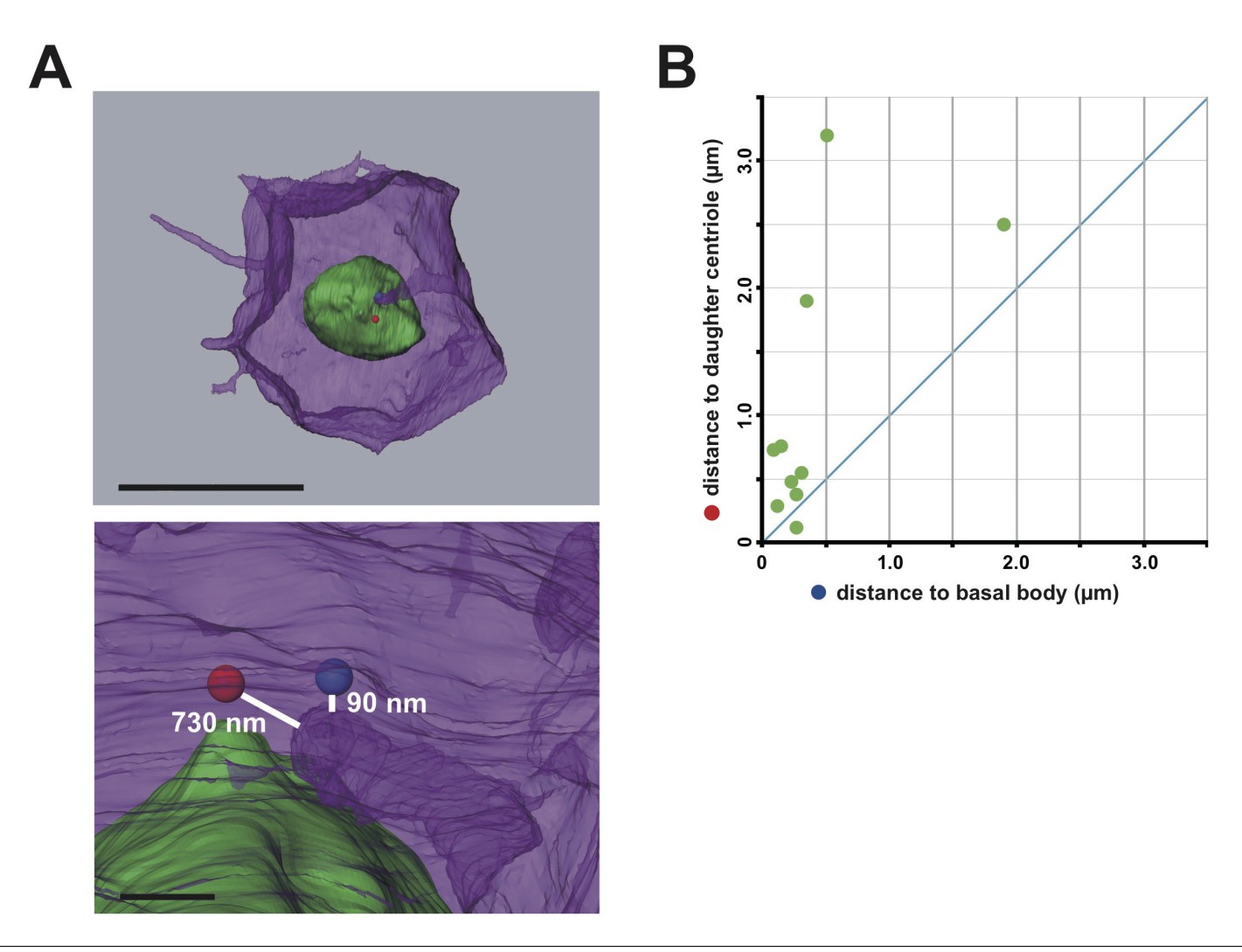

**Figure 6.** The membrane invagination approaches closer to the basal body than to the daughter centriole. (**A**) A representative segmented cell from SBF-SEM data. Green: nucleus. Blue and red balls: the basal body and daughter centriole, respectively. Upper panel: top view of the whole cell. Bar: 10 μm. The lower panel shows a closer view at the tip of the invagination. Distances between 3D objects were measured in AMIRA. We recognized the mother centrioles/basal bodies by the presence of cilia in the serial sections of SBF-SEM. (**B**) Distance of the tip of the individual invagination to the basal body or daughter centriole. The X- and Y-axis show the distance to the basal body and daughter centriole, respectively. We measured ten invaginations and found that nine of ten invaginations approached the basal body more closely than the daughter centriole.

## Membrane invagination depends on microtubules and are under tensile force

Next, we investigated the involvement of the cytoskeleton in forming the invaginations. Microtubule depolymerisation with nocodazole blocked the formation of invaginations completely (*Figure 7B*), indicating that MTOC activity induces the invaginations and that, in turn, the invaginations relocate the MTOC to the posterior of cells. On the other hand, a treatment of embryos with cytochalasin, which perturbs actin polymerisation, increased the number of invaginations, which now originated from all lateral plasma membrane domains (*Figure 7A–C* and *Video 10*). Thus, the tubular membrane structure of ascidian epidermal cells depends on microtubule function, reminiscent of the 'nanotubes' described in *Drosophila* male germ cells (*Inaba et al., 2015b*), rather than the actin-based membrane tubes described in other systems (ex. some cytonemes and tunneling nanotubes) (*Gerdes and Carvalho, 2008*; *Hsiung et al., 2005*; *Rustom et al., 2004*; *Sanders et al., 2013*).

*Drosophila* microtubule-based nanotubes (MT-nanotubes) are filled with microtubules (*Inaba et al., 2015b*). During the course of live imaging analyses, we did not observe microtubules within the membrane invaginations. By contrast, we often observed that the plus end of the microtubule labeled with EB3-mCherry reached to the tip of the invagination (*Figure 7D*). Further, we explored the topological relationship between the membrane invagination and microtubules by serial transmission electron microscopy (serial-TEM) observation (*Figure 7E*, *Video 11*). In serial-TEM, the membrane invaginations were visualized as double bilayer fragments (*Figure 7Ea–f*) extending toward the basal body (*Figure 7Eh–k*). Abundant microtubules were also observed between the tip of invagination and the basal body (*Figure 7Ef–h*), but no

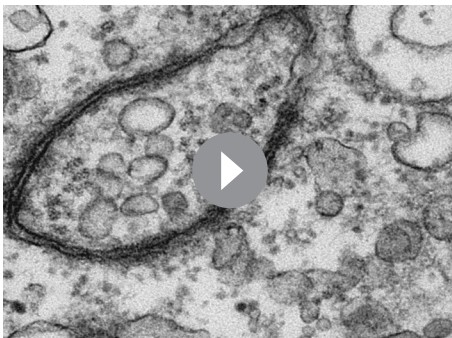

**Video 11.** A video made of sections of serial-TEM images shows the invagination toward the basal body. This video is related to *Figure 7E*.

microtubules were observed within the invagination itself. These results indicate that the membrane invagination in ascidian epidermal cell is distinct from microtubule-filled nanotubes although both tubular structures are sensitive to microtubule depolymerization. Our results also indicate that the membrane invagination may associate with the centrosome via microtubules.

Based on these observations, we speculated that the membrane invagination is physically connected to the centrosome and may have a role in positioning it to the posterior side of the cell. To examine whether a pulling force was generated between the membrane invaginations and the posteriorly located centrioles, we employed laser ablation (*Rauzi et al., 2008*). The junction of the fluorescently labelled invagination and the EB3-mCherry-labelled centrosome was cut by UV laser irradiation, and the cells were observed by time-lapse imaging (*Figure 8A*, white cross). Interestingly, the end of the cut edge of the invaginations regressed immediately to the basal position, suggesting that the invaginations are normally under tension (*Figure 8A*, *Video 12*). Therefore, the displacement of the nucleus during the 11th cell-division cycle could be explained by a posteriorly directed force (*Figure 1D,E*). In the series of laser ablation experiments (*Figure 8B*), we noted that

the longer invaginations had a tendency to regress more rapidly (*Figure 8C*). The results show a positive correlation between the initial length and both average and max speed of regression (recoiling). When we cut one of two invaginations that cooperatively support a single centrosome, a significant recoil occurred toward the intact (uncut) invagination (*Video 13*). This finding strongly suggests that the centrosome was balanced by two invaginations that generate pulling forces. Interestingly, soon (15 min) after the cutting, we found that the plasma membrane re-invaginated toward the centrosome (*Figure 8D*).

## The orientation of cell division is correlated with the directionality of the membrane invagination

The regeneration potential of the membrane invagination obstructed the functional analysis by laser ablation. Thus, we decided to indirectly alter the direction of invaginations by disruption of the planar cell polarity (PCP) pathway, which establishes polarity in multicellular tissues

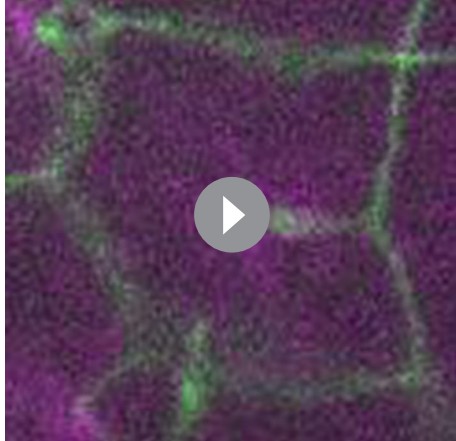

**Video 12.** (10 fps) A time-lapse movie of an embryo expressing PH-GFP (green) and EB3-mCherry (magenta), made with a time series of a single confocal plane. Laser irradiation occurs at the third frame. Ten seconds are compressed to one second. Anterior: left. This video is related to *Figure 8A*.

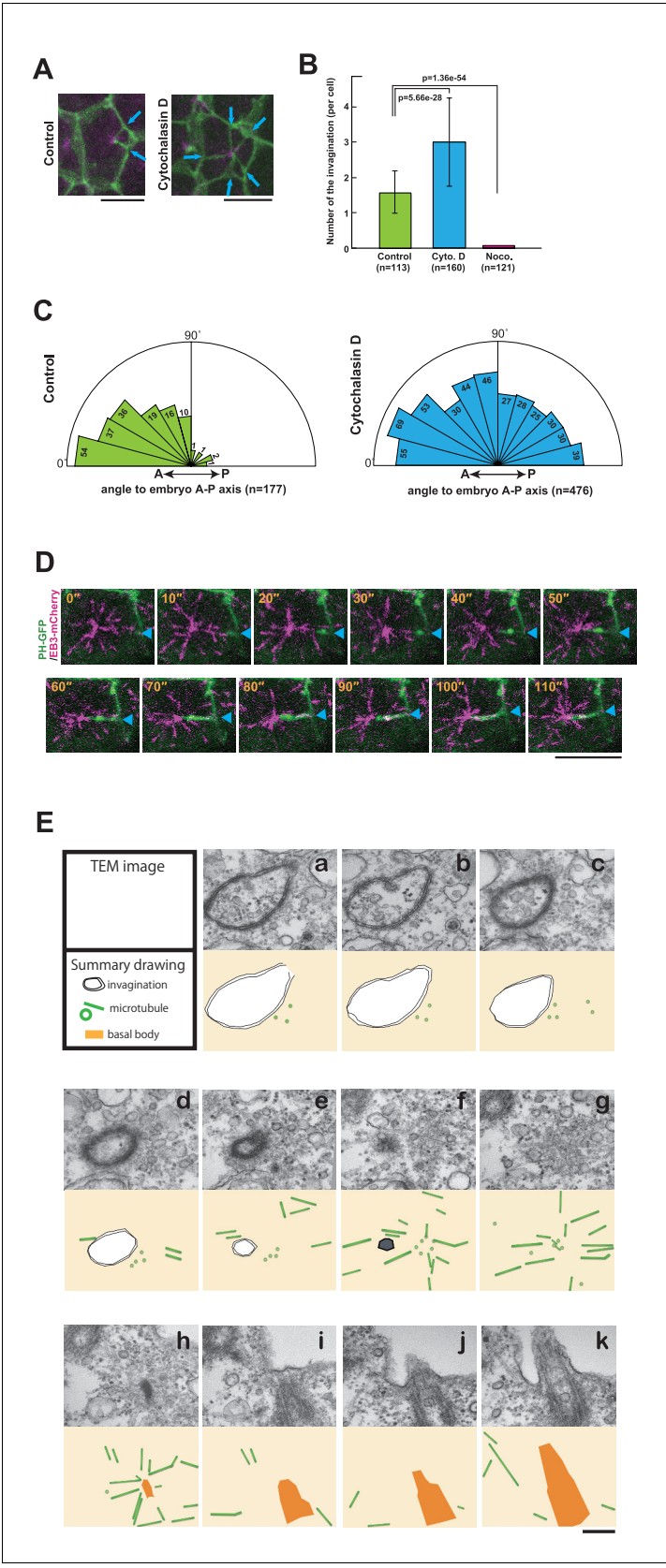

**Figure 7.** The role of cytoskeletal elements in the formation of the membrane invagination and the implication of microtubule function. (**A**) Representative epidermal cells expressing PH-GFP/EB3-mCherry in control and

*Figure 7 continued on next page*

*Figure 7 continued*

cytochalasin-treated embryos. The cytochalasin panel is from *Video 10*. Blue arrows: membrane invaginations. Bars: 10 µm. Anterior: left. (**B**) The number of invaginations after inhibitor treatment: cytochalasin treatment (160 cells from three embryos) increased the number of invaginations, while nocodazole treatment (121 cells from three embryos) decreased the number of invaginations compared to control (113 cells from three embryos). Histograms are presented as the mean ± SD. p-values were obtained using the Welch's t-test. (**C**) In cytochalasin-treated cells, unlike normal cells, invaginations also formed from the anterior side and extended toward the posterior of the cell. We counted 177 and 476 invaginations, respectively, in three control and three cytochalasin-treated embryos. (**D**) A high time-resolution timelapse recording of a representative cell expressing PH-GFP/EB3-mCherry. A blue arrowhead indicates the membrane invagination. Time elapsed from the start of recording is shown in orange. Bar: 10 µm. Anterior: left. (**E**) A series of images from the serial TEM observation. TEM images (upper panels) and the corresponding schematic drawings (lower panels) are shown with the microtubules as green tubes, the membrane invagination as black double-line in a–e, or its tip as a hexagon filled by gray in f, and the basal body as an orange structure in h–k.

(*Adler, 2002*; *Gubb and Garcia-Bellido, 1982*; *Wallingford, 2012*). To disrupt the PCP pathway, we depleted Dishevelled (Dsh) (*Hotta et al., 2003*; *Theisen et al., 1994*; *Wallingford and Habas, 2005*), a core component of PCP, by injection of antisense morpholino oligo (MO). This resulted in a radial formation of multiple membrane invaginations (*Figure 9A,B*, *Figure 10A*, blue arrows, and *Video 14*). Disruption of directional membrane invaginations was strongly correlated with randomization of mitotic spindle orientation (*Figure 9A,B*, *Figure 10A* and *Video 14*). Co-injection of *Dsh* mRNA with *Dsh* MO rescued both the direction of invaginations and cell division orientation along the A-P axis (*Figure 9C*). These results suggest that the posterior distribution of the invaginations is tightly associated with the direction of mitotic divisions. We also found that the nuclei were distributed around the center of the cells with the omni-directional invaginations, as opposed to the posterior localisation observed in the cells with directional invaginations (*Figure 10B*). Moreover, the centrosomes were also positioned in the center of the cells with the radial formation of invaginations (*Figure 10A*; Omni-directional 86', *Figure 10C* and *Video 14*). Following this, the two duplicated centrosomes exhibited equivalent motility in the

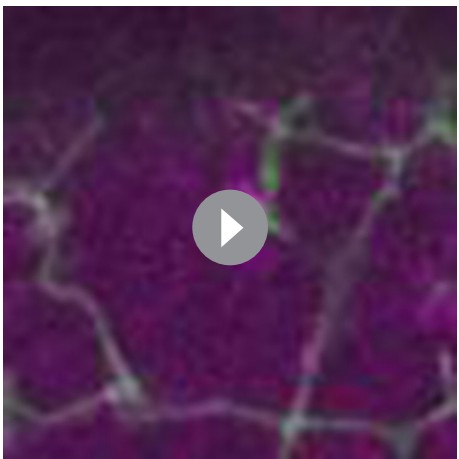

**Video 13.** (10 fps) A time-lapse movie of an embryo expressing PH-GFP (green) and EB3-mCherry (magenta), made with a time series of a single confocal plane. Laser irradiation on the lower invaginations occurs at the 10th frame. This clearly shows a bounce toward the intact invagination after the ablation of the other. Ten seconds are compressed to one second. Anterior: left. This video is related to *Figure 8A*.

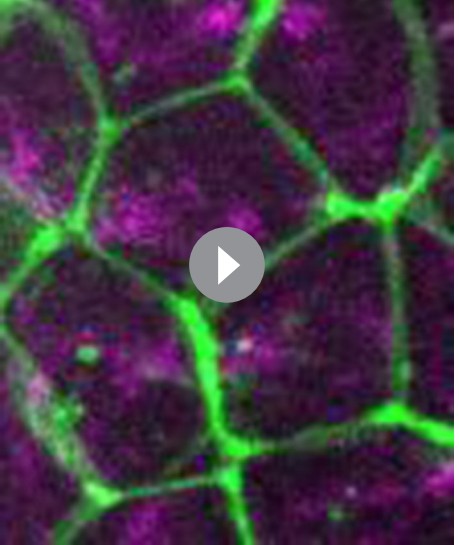

**Video 14.** (10 fps) A time-lapse movie of an embryo expressing PH-GFP (green) and EB3-mCherry (magenta), made with the maximum-intensity projection of the confocal data. This shows that the radially formed invatginations affected the centrosome dynamics. Five minutes are compressed to one second. Anterior is left. This video is related to *Figure 10A* (lower panels).

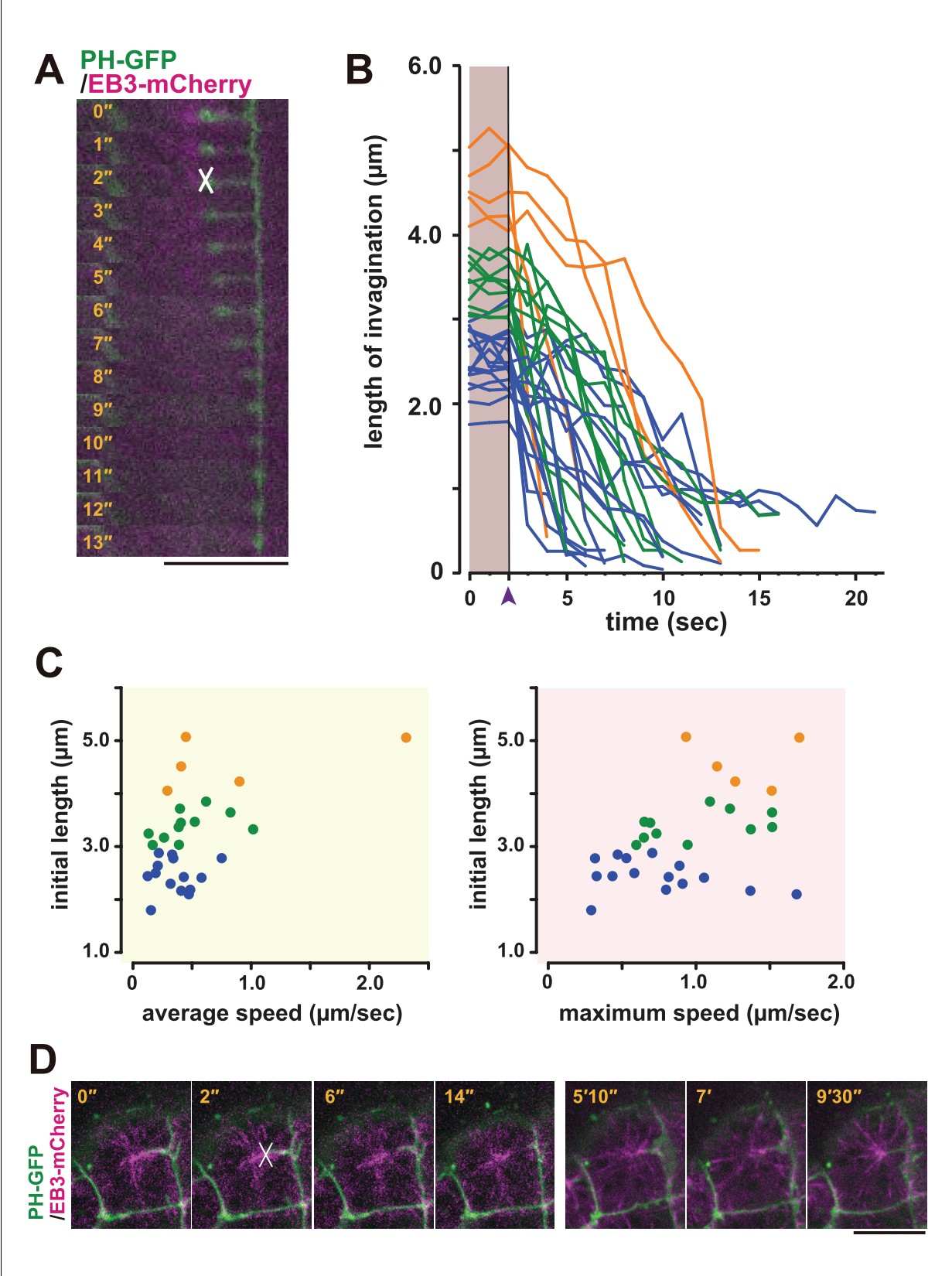

**Figure 8.** Laser ablation and regeneration of the membrane invagination. (**A, B**) Effect of UV laser ablation on membrane invagination in a cell expressing PH-GFP/EB3-mCherry. (**A**) Image shows frames from *Video 12*, with the elapsed time indicated. In the 2'' frame, the white 'X' indicates the

*Figure 8 continued on next page*

*Figure 8 continued*

point of laser ablation. The membrane recoiled rapidly after ablation. Bar: 10 μm. (B) Graph showing measurements of invagination length following laser ablation. The black line, indicated with purple arrow head, shows the time of ablation. A colour code was used to highlight the difference of initial length of the membrane invaginations when the laser ablation was conducted: orange, >4 μm; green, 4 μm to 3 μm; blue, <3 μm. n = 31 invaginations. (C) The correlation between the initial length and average (left) or maximum (right) speed. Coloured dots correspond to the lines in B. The correlation coefficient (r) is 0.52 or 0.44 for the average speed or the maximum speed, respectively (Pearson correlation). (D) The plasma membrane re-invaginated after UV laser ablation. We observed this regeneration event at least five independent experiments. The white 'X' indicates the point of laser ablation. Time elapsed from the start of recording is shown in orange. Frames for 0'-14' correspond to single confocal planes while those for 5'10'–9'30' are max intensity projections. Bars: 10 μm. Anterior: left.

cells with the omni-directional invaginations (*Figure 10A*; Omni-directional 105', *Figure 10D* and *Video 14*), in contrast to the asymmetric centrosome migration found in the cells displaying directional invaginations (*Figure 10A*; Directional 83', *Figure 10D*). Finally, epidermal cilia form in the center of the cells with omni-directional invaginations, instead of at a posterior position (*Figure 10E, F*). These results suggest that the centrosome dynamics in ascidian epidermal cells is tightly correlated with the direction of membrane invaginations.

## Discussion

We report here that an unusual membrane structure, which forms during interphase of a specific embryonic cell cycle, is correlated with the dynamics of centrosomes. This includes positioning of centrosomes and cilia and determination of the cell division axis (graphical summary and model in *Figure 11A,B*). We propose a model whereby these events are coupled by physical interaction between a centriole and plasma membrane, which results in the generation of a tensile force in the latter. Although we could not directly demonstrate that the membrane invagination is required for mitotic spindle orientation, our data is consistent with our model whereby the centrosome is captured by a unique membrane structure and is subsequently tethered to the posterior side of cells. In the *Drosophila* stem cell systems, duplicated centrosomes generated from an eccentrically positioned centrosome show differential motilities (*Yamashita and Fuller, 2008*). In the ascidian epidermal cells in the 11th cell cycle, the duplicated centrosomes also exhibit differential motilities with the less motile centrosome remaining associated with the membrane invagination. Although some microtubule (MT)-filled nanotubes resist nocodazole treatment (*Önfelt et al., 2006*), the formation of the ascidian membrane invagination, like the MT-nanotubes in *Drosophila* male germ cells (*Inaba et al., 2015b*), depends on microtubule function. However, the distribution of the microtubule is distinct in the latter two tubular membrane structures and it is not known whether MT-nanotubes protrude toward the centrosome. Interestingly, a similar membrane invagination to the one we observed in ascidian embryos was previously reported in the *C. elegans* zygote, and it was suggested that the actomyosin cortex counteracts pulling forces mediated by microtubules, resulting in the posterior displacement of the spindle and the unequal cell division (*Redemann et al., 2010*). Our pharmacological tests with either microtubule or actin inhibitors resulted in similar observations to those described for the *C. elegans* membrane invagination. Importantly, we report here for the first time that membrane invaginations take place in normal (untreated) cells in a multicellular tissue (ascidian embryonic epidermis). Our observations collectively indicate that these invaginations (generally one or two per cell) arising from two adjacent plasma membranes may control centrosome dynamics and the orientation of the cell division axis. It should be also noted that a similar membrane invagination is observed during the formation of immunological synapses between T cells and antigen-presenting cells. During this process, T cells rapidly reposition their centrosome to the center of the immunological synapse. Importantly, the centrosome repositioning is coupled with the formation of a membrane invagination originating from the synaptic interphase and reaching towards the T-cell centrosome in a mirotubule-dependent manner (*Yi et al., 2013*). Therefore, the microtubule-dependent mechanism in which a membrane invagination captures and repositions the centrosome may be a general mechanism that transcends species and cell types for polarised centrosome repositioning. In ascidian epidermal cells, this process appears to be also involved in the posterior positioning of the cilia. In this study, we revealed that the tip of the membrane invagination is found in close vicinity of the basal body, indicating that the membrane

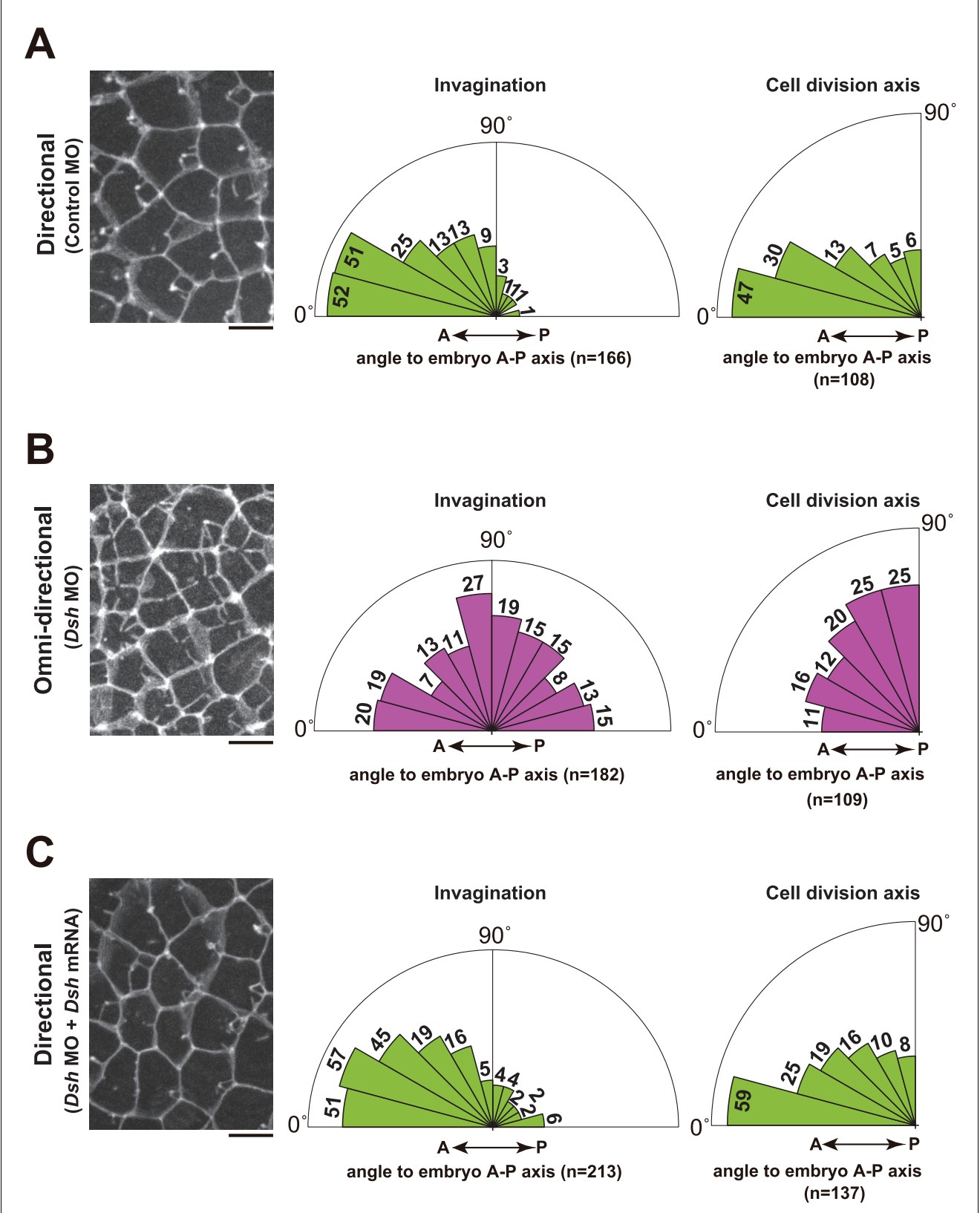

**Figure 9.** Depletion of Dsh resulted in the radial formation of multiple membrane invaginations and randomized the cell division axis in the 11th cell cycle epidermal cells. (**A**) The directional invaginations labeled by PH-GFP in the control MO-injected cells. The bar represents 10 µm (left). Rose

*Figure 9 continued on next page*

*Figure 9 continued*

diagrams showing the angle of the invagination relative to the embryonic A–P axis, n = 166 invaginations from three embryos (middle), and the angle of cell division relative to the embryonic A–P axis, n = 108 cells from three embryos (right). The results are almost same as the normal embryos in *Figure 1* and *Figure 2*. (B) The omini-directional invaginations labeled by PH-GFP in the *Dsh* MO-injected cells. The bar represents 10 μm (left). Rose diagrams showing the angle of the invagination relative to the embryonic A-P axis, n = 182 invaginations from three embryos (middle), and the angle of cell division relative to the embryonic A-P axis, n = 109 cells from three embryos (right). In the embryos with radially formed invaginations, the orientation of cell division is randomized. (C) Co-injection of *Dsh* MO and *Dsh* mRNA restores directional invaginations. The bar represents 10 μm (left). Rose diagrams showing the angle of the invagination relative to the embryonic A-P axis, n = 213 invaginations from three embryos (middle), and the angle of cell division relative to the embryonic A–P axis, n = 137 cells from three embryos (right). The results are reminiscent of normal and the control MO-injected embryos.

invagination might associate preferentially with the mother centriole. Since the mother and daughter centrioles inherit distinct components (*Pelletier and Yamashita, 2012*; *Pereira et al., 2001*), specific molecules of the mother centriole might be involved in establishing the proposed association with the membrane invagination. Posteriorly localized cilia are found in several systems and it will be interesting to determine if their position depends on similar mechanism (*Antic et al., 2010*; *Borovina et al., 2010*; *Hashimoto et al., 2010*; *Momose et al., 2012*; *Thompson et al., 2012*).

In this study, we found the positive correlation between the initial length of the membrane invagination and both the average and maximum speed of regression following laser ablation. This observation implies that the membrane invagination has an elastic restoring force rather than an active force generator like cell-cell junctions (*Ishihara and Sugimura, 2012*; *Rauzi et al., 2008*).

Some questions remain to be addressed—for example, which molecule(s) acts as the force generator to invaginate the plasma membrane and establish a link with the centrosome. The microtubule-dependent invagination process targeting the centrosome implies the involvement of plus-end-directed motor. Indeed, the generation of membrane invaginations in *C. elegans* one-cell embryos involves cortical dynein motors (*Redemann et al., 2010*). Therefore, this plus-end motor protein would be one of the promising candidates for the formation of invagination in ascidian epidermal cells. The rigidity of the plasma membrane (*Ramanathan et al., 2015*; *Stewart et al., 2011*) should be measured during the cell cycle to clarify the mechanism of the phase transition of the membrane invagination, from 'approaching' to 'pulling' the centrosome. Furthermore, by disrupting a core PCP component in ascidian epidermal cells, we have shown a tight correlation between the direction of membrane invagination and centrosome dynamics including mitotic spindle orientation. However, in this experiment, we cannot rule out the possibility that the PCP pathway affects centrosome behavior independently of this membrane structure. Hence, it should be further addressed how the PCP pathway participates in the regulation of centrosome dynamics and the polarization of the membrane invagination. It will also be intriguing to investigate whether the PCP pathway regulates the asymmetric distribution of the actomyosin network and/or plus-end motors in ascidian epidermal cells. Nevertheless, our present work uncovers a previously unknown mechanism associating the centrosome and the plasma membrane, and may open new avenues of investigation into the hidden mechanisms of oriented cell division that underlie embryogenesis and organogenesis.

## Materials and methods

### Embryo handling

*Ciona intestinalis* adults were supplied by the National Bio Resource Project (NBRP, Japan) or purchased from the Station Biologique de Roscoff (France). Eggs, embryos, and microinjections were handled following conventional protocols (*Sardet et al., 2011*). We injected mRNAs into unfertilized eggs or eight cell stage embryos. The embryos were cultured and observed at 20℃. For inhibitor treatment, embryos that had just reached the 10th cell division were placed in artificial seawater methylcellulose (ASWM) containing nocodazole (1 nM final concentration; 1:200,000 dilution of DMSO stock) or cytochalasin (10 μg/ml final concentration; 1:1000 dilution of ethanol stock), or 0.1% ethanol (control).

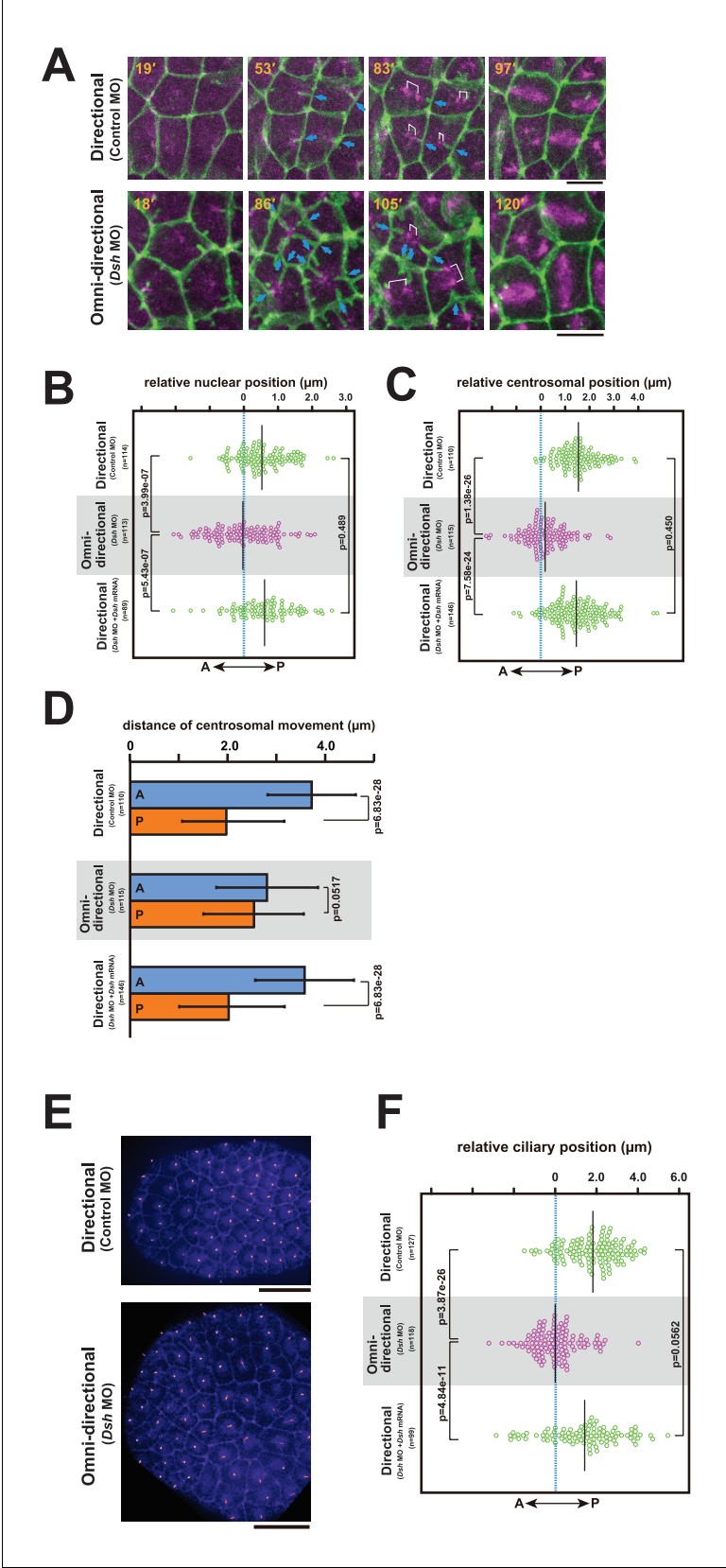

**Figure 10.** The centrosome dynamics is highly correlated with the directionality of the membrane invagination. (**A**) Centrosome dynamics with the directional (upper) or the omni-directional (lower; frames from *Video 14*) membrane invaginations in embryos injected with MOs and expressing PH-
*Figure 10 continued on next page*

*Figure 10 continued*

GFP/EB3-mCherry. Blue arrows indicate the invaginations. Elapsed time is indicated in each panel. Anterior is left. Bars: 10 μm. (B) Bee-swarm plots showing the nuclear position relative to the centre of the cell along the embryonic A–P axis with the directional (upper; control MO, n = 114 cells from three embryos and lower; *Dsh* MO + *Dsh* mRNA, n = 89 cells from three embryos) and the omini-directional (middle; *Dsh* MO, n = 113 cells from three embryos) invaginations. Black lines show the average nuclear position relative to the centre of the cell (blue dotted line); the average position was 0.53, −0.03 and 0.6 μm toward the posterior side in the embryos injected with control MO, *Dsh* MO and *Dsh* MO + *Dsh* mRNA, respectively. p-values were obtained using the Welch's t-test. (C) Centrosome position relative to the centre of the cell along the embryonic A–P axis just before duplication with the directional (upper; control MO, n = 110 cells from three embryos and lower; *Dsh* MO + *Dsh* mRNA, n = 146 cells from three embryos) and the omini-directional (middle; *Dsh* MO, n = 115 cells from three embryos) invaginations. Black lines show the average centrosome position relative to the centre of the cell (blue dotted line); the average centrosome position was 1.54, 0.18 and 1.46 μm toward the posterior side in the embryos injected with control MO, *Dsh* MO and *Dsh* MO + *Dsh* mRNA, respectively. p-values were obtained using the Welch's t-test. (D) Histograms showing migration distance of anterior and posterior centrosomes over the period from centrosome duplication to the end of migration in epidermal cells with the directional (upper; control MO, n = 110 cells from three embryos and lower; *Dsh* MO + *Dsh* mRNA, n = 146 cells from three embryos) and the omini-directional (middle; *Dsh* MO, n = 115 cells from three embryos) invaginations. Blue and orange columns show the migration of the anterior and posterior centrosomes, respectively. The data are presented as the mean ± SD. p-values were obtained using the Welch's t-test. (E) Cilium positioning in epidermal cells with the directional (upper) or the omini-directional (lower) membrane invaginations in embryos expressing Arl13b-GFP. (F) Bee-swarm plots showing cilium position relative to the centre of the cell along the embryonic A-P axis in epidermal cells with the directional (upper; control MO, n = 127 cells from three embryos and lower; *Dsh* MO + *Dsh* mRNA, n = 99 cells from three embryos) and the omini-directional (middle; *Dsh* MO, n = 118 cells from three embryos) invaginations. Black lines show the average cilium position relative to the centre of the cell (blue dotted line); the average cilium positions were 1.82, 0.00 and 1.42 μm toward the posterior side in embryos injected with control MO, *Dsh* MO and *Dsh* MO + *Dsh* mRNA, respectively. p-values were obtained using the Welch's t-test.

## Constructs and MO

PH-GFP was a gift from Dr. A. McDougall; H2B-mCherry was a gift from Dr. H. Nishida. We constructed pRN3-membraneGFP (*Morita et al., 2012*) by PCR-amplifying the ORF from pCS2+ membraneGFP (forward primer CAACTTTGGCAGATCTGGATCCCATCGATTCGAA; reverse primer GCCCTATAGTGAGTCGTATTACGTAGCGGCCGCGGATCTGGT) and subcloning it into the pRN3 vector with an In-Fusion HD cloning system (Clontech, Mountain View, CA). The pRN3 was digested by BglII and NotI. To construct PH-tdTomato, we made pRN3-RfA-tdTomato from pCX3-RfA-tdTomato, a gift from Dr. T. Momose. The pCX3-RfA-tdTomato was digested by Acc65I and blunt-ended. RfA-tdTomato was cut out by the restriction enzyme BglII and subcloned into the pRN3 vector. The ORF of the PH domain was PCR-amplified from PH-GFP (forward primer AAAGGATCCACCATGGACTCGGGCCGGGACTTCCT; reverse primer TTTGAATTCCCCGGGGGATGTTGAGCTCCTTCAGGA) and subcloned into the pENTR vector, which was mixed with pRN3-RfA-tdTomato in a Gateway LR reaction (Life Technologies, Carlsbad, CA). We constructed EB3-mCherry from EB3-GFP, a gift from Dr. A. Akhmanova. The EB3-GFP was digested by EcoRI and XhoI and subcloned into pRN3 between the EcoRI and NotI sites to make pRN3-EB3-GFP. Next, the mCherry ORF was removed from H2B-mCherry by BamHI digestion and inserted into the BamHI site of pRN3-EB3-GFP. To construct ensconsin-tdTomato, the ensconsin ORF was amplified by PCR (forward primer CAACTTTGGCAGATCTACCATGGAGCAGAAGCTCATCTC; reverse primer CTTGCTCACCATGATATCGACCGGTGGATCCGAAGA) from pHTB-ensconsin-3xVenus (*Negishi et al., 2013*) and subcloned with an In-Fusion HD cloning system into pRN3-tdTomato, which was created by removing the PH domain from pRN3-PH-tdTomato by BglII and EcoRV. To construct pRN3-Arl13b-GFP, the *Ciona* Arl13b (NCBI accession number: XP_002129357) ORF was cloned by PCR (forward primer CAACTTTGGCAGATCTACCATGATCGGTCAAATGGGG; reverse primer CATGAATTCAGATCTAACAA-CAATGTCTTCCTCAGAAT) from a cDNA library (a gift from Dr. Miho Suzuki), and was inserted with the In-Fusion HD cloning system into pRN3-GFP, which was created by digesting the PH domain out from pRN3-PH-GFP by BglII. *Ciona Dsh* (Aniseed Gene model: KH2012:KH.L141.37) MO (5'-AACAATTTTCGTTTCATCCGACATT-3', Gene Tools, Philomath, OR) and standard control MO (Gene Tools, Philomath, OR) were injected at 0.3 mM. The rescuing Dsh construct was generated by changing the nucleotides at the MO target site as follows, (−1) 5'-catgGAgACaAAgATcGTgTAT-3' (+24), small letters indicate the replaced nucleotides with PCR (forward primer CAACTTTGGCAGATCTACCATGGAGACAAAGATCGTGTATTATCTTGGCGATGAACAAA; reverse primer ACCAGATCCGCGGCCCATGACGTCAACAAAATAATCAC). *Dsh* mRNA was injected at 1.0 μg/μl. The mRNAs were synthesised using an mMESSAGE mMACHINE kit (Life Technologies, Carlsbad, CA).

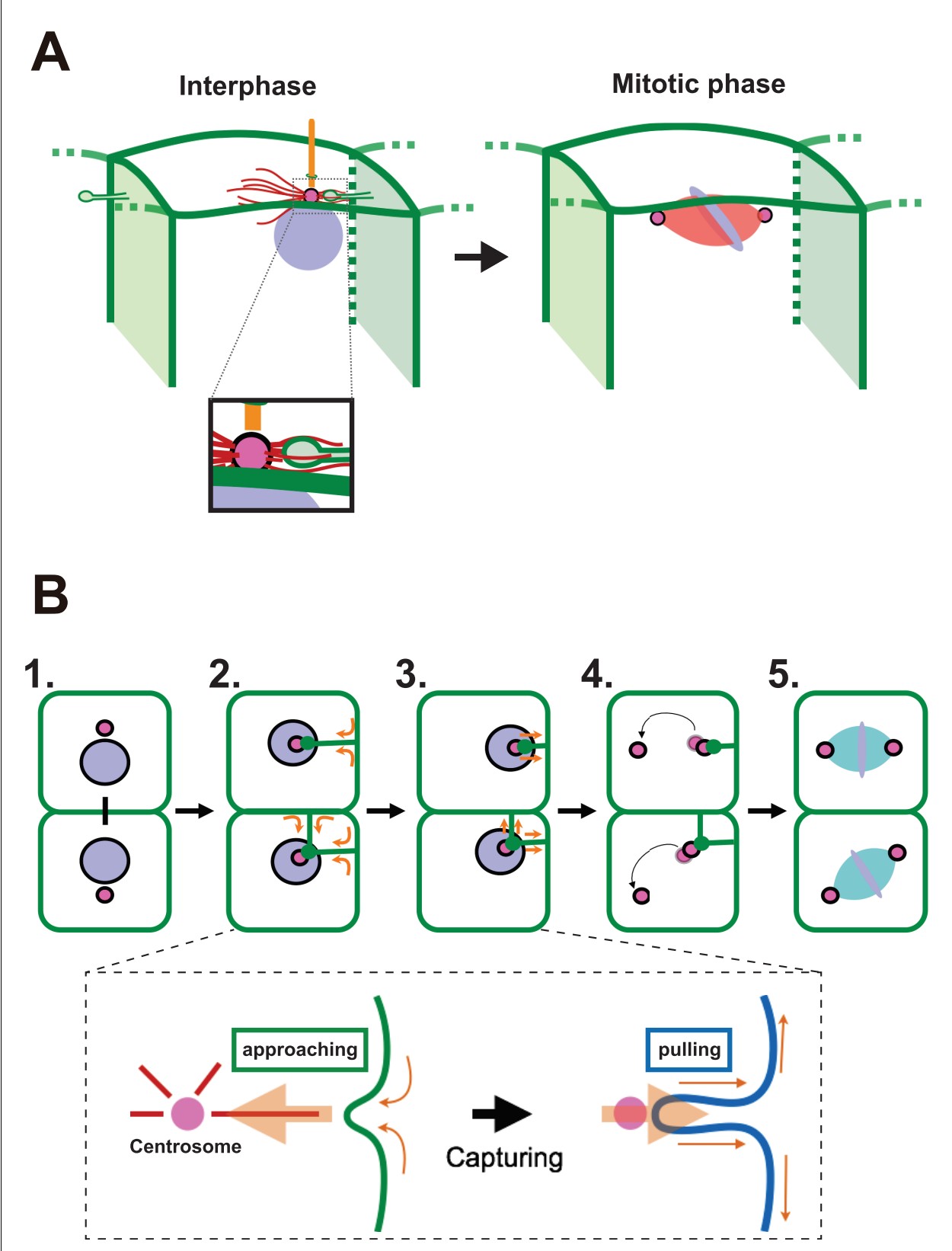

**Figure 11.** A proposed model for mitotic spindle orientation driven by membrane invaginations. (**A**) Graphical summary of the spatial relationship of each component. Anterior is left. Interphase: nuclei (purple), cell membrane (green), centrosome (magenta), microtubule (red) and cilium (orange). *Figure 11 continued on next page*

*Figure 11 continued*

Mitotic phase: neither invaginations nor cilia are found. (**B**) Schematic drawing of the course of 11th epidermal cell division in relation to the invagination. Apical view of cells. Upper panel: (1) Nuclei (purple) and centrosomes (magenta) just after the 10th cytokinesis; (2) After the centrosome migrates toward the apical cortex, the membrane invaginates toward the centrosome from the posterior plasma membrane; (3) The invagination shrinks, pulling the centrosome and nucleus toward the posterior; (4) The centrosomes duplicate and exhibit distinct migratory activities with the less motile centrosome remaining associated with the membrane invagination; (5) The mitotic spindle forms aligned along the A-P axis. In the lower panel, the assumed forces involved in the invagination are shown. MTOC activity on the centrosome causes the 'approaching' invagination from the plasma membrane, likely to be depending on microtubule function. After the centrosome is associated with the tip of membrane invagination, a tensile force acting on the invagination brings the centrosome toward the posterior.

## Imaging

Ascidian embryos were prepared for live imaging as described previously (*Negishi et al., 2013*). We injected mRNAs as follows; PH-GFP (1.8 µg/µl), PH-tdTomato (2.0 µg/µl), H2B-mCherry (1.2 µg/µl), EB3-GFP (2.0 µg/µl), EB3-mCherry (2.5 µg/µl), membraneGFP (2.0 µg/µl), and ensconsin-tdTomato (2.5 µg/µl). For FM4-64 (Life Technologies) staining, the embryo was incubated in artificial seawater containing the dye (10 µg/ml) for three hours, and then the stained embryo was mounted in ASWM.

The embryos were observed with an Olympus IX 81 (60x / 1.20 NA water immersion lens, Olympus, Japan) with a Yokogawa CSU-X1 spinning-disk confocal unit and an iXon3 897 EM-CCD camera (Andor, UK). For laser ablation, an $N_2$ Micropoint laser (16 Hz, 365 nm wave length, Photonic Instruments, US, Arlington Heights, IL) attached to a CSU microscopy system was used. Images were obtained with Andor IQ2 software. Embryos were also observed with a Leica SP5 (40x / 1.30 NA oil immersion lens) or Leica SP8 (63x/1.20 NA water immersion lens), or Nikon A1 (60x / 1.20 NA water immersion lens) scanning confocal microscopy system.

## Immunostaining

Acetylated tubulin immunofluorescence was previously described (*Hudson and Lemaire, 2001*). The stained embryos were mounted in Fluoro-KEEPER with DAPI (Nacalai Tesque, Japan).

## Quantification

Acquired data were processed with ImageJ software (http://imagej.nih.gov/ij/). The angle, length, and centroid of the cell and the nucleus were manually measured with ImageJ. The length and direction of the invagination were measured in the maximum-intensity projection of time-lapse movies of embryos expressing PH-GFP at the time point when an invagination reached its maximum length (*Figure 1B*, *Figure 2E–G* and *Figure 7C*). The cell division axis relative to the embryonic A-P axis was also measured in the maximum-intensity projection of time-lapse movies of embryos expressing PH-GFP (*Figure 1B* and *Figure 9*). The long axis of the cell was determined and measured just before the nuclear membrane breakdown, and we calculated the angle of the cell division axis relative to the long axis (*Figure 1C*). In embryos expressing PH-GFP/H2B-mCherry, we measured the position of individual nuclei just before the nuclear membrane breakdown. We measured the centroid of the nucleus and the cell, and calculated the distance of both centroids along the A-P axis (*Figure 1E* and *Figure 10B*). We also measured the centrosome position just before the separation of the duplicated centrosomes during the 10th and 11th cell division cycles in embryos expressing PH-GFP/EB3-mCherry, and normalized the centrosome position to the centroid of the cell (*Figure 4D* and *Figure 10C*). To measure the migration distance of duplicated centrosomes, we calculated for each centrosome the distance between the position just before migration started and that when migration ceased in embryos expressing PH-GFP/EB3-mCherry; both positions were normalized to the centroid of the cell (*Figure 4E* and *Figure 10D*). We also used embryos expressing PH-GFP/EB3-mCherry to measure the angle of the centrosome axis relative to the invagination axis; the line between the two duplicated centrosomes just after their separation during the 11th division cycle was used as the centrosome axis (*Figure 4F*). When there were multiple invaginations, we calculated the composition of these structures as a vector (see also *Figure 4F*). To quantify the cilium's position in DAPI/acetylated tubulin-immunostained embryos, we traced the cell contours, calculated the centre of the cell, and then measured the distance between the cilium and the centre of the cell along the A–P axis (*Figure 5C*). We also quantified cilia positions at 30 min before the 11th

cytokinesis started with Arl13b-GFP labelled cilia in MO-injected embryos. In the laser ablation experiments, we measured the length of invagination in each frame from a single confocal plane for more than 10 min (*Figure 8*). To calculate the average speed of regression after UV laser ablation, we divided the length of invagination by the total time and in the case of maximum speed, we measured the change of length between each time frame (one second). All measurements were completed manually with ImageJ (*Schneider et al., 2012*). All results showing statistically significant difference were supported by power analysis with R-package compute.es (http://cran.r-project.org/web/packages/compute.es) and pwr (https://cran.r-project.org/web/packages/pwr). All data were obtained from at least three different batches of embryos.

## Sample preparation for SBF-SEM

*Ciona intestinalis* tailbud embryos were washed with washing buffer [50% artificial seawater containing 0.1 M sodium cacodylate buffer (pH 7.4)] and fixed in prefix buffer [2.5% glutaraldehyde and 2.0% paraformaldehyde in 0.1 M sodium cacodylate buffer (pH 7.4) and 50% artificial seawater] overnight at 4°C. The prefixed embryos were then triple-washed with washing buffer and postfixed with 2% osmium tetroxide and 1.5% potassium ferrocyanide in 0.1 M sodium cacodylate (pH 7.4) and 50% artificial seawater for 1 hr at 4°C. The samples were then washed with distilled water, incubated with 1% thiocarbohydrazide in distilled water for 1 hr at 60°C, washed again with distilled water, and postfixed again with 2% osmium tetroxide in distilled water for 30 min at room temperature. After washing with distilled water, the fixed embryos were stained *en bloc* with 1% uranyl acetate in distilled water overnight at 4°C, and then with 0.2 M lead aspartate (pH 5.5) for 30 min at 60°C. The embryos were washed with distilled water and dehydrated by a graded series of ethanol (50–100%) at 4°C and acetone at room temperature. Finally, they were infiltrated with durcupan resin, and the resin was polymerized at 60°C for 3 days.

The resin blocks containing the embryos were manually trimmed with a razor blade and glued onto an aluminum SBF-SEM rivet with conductive epoxy resin (SPI Conductive Silver Epoxy; SPI Supplies and Structure Prove, Inc., West Chester, PA, USA). Specimens on the rivet were further trimmed with a razor blade to as small a size as possible to include only one embryo (about 100–200 μm). All lateral surfaces of the specimens were ion-coated with gold to a thickness of 20 nm to dissipate the electric charge caused by electron-beam irradiation during SEM imaging.

## Acquisition of image stacks using SBF-SEM and image analysis

In this study, we used a system originally developed at the Max Planck Institute for Medical Research, Heidelberg, Germany (*Denk and Horstmann, 2004*); in this system, a scanning electron microscope (MERLIN, Carl Zeiss Microscopy, Jena, Germany) equipped with an in-chamber ultramicrotome system (3View; Gatn Inc., Pleasanton, CA, USA) was used for slicing and imaging the SBF-SEM stacks. Specimens mounted on SBF-SEM rivets were placed on the specimen stage in the SEM chamber, and the block surface was sliced by the in-chamber ultramicrotome and imaged each time with a back-scattered electron detector. The serial image stacks were acquired automatically as reported previously (*Miyazaki et al., 2014*). The SBF-SEM images were recorded with an accelerating voltage of 1.5 kV with a dwell time of 1.0 μs. The image size was 8192 × 8192 pixels. The slice thickness was 50 nm. After 2× binning of the images, the image stack was automatically aligned using 'Register Virtual Stack Slices' in the Fiji/ImageJ software package (http://fiji.sc/Fiji) (*Schindelin et al., 2012*). Individual cells and organelle structures were manually segmented using the AMIRA software package (FEI Visualization Science Group, Burlington, MA, USA). This software package was also used to generate the figures.

## Sample preparation for serial transmission electron microscopy (serial-TEM)

*Ciona* tailbud embryos were fixed in prefix buffer [2% gultaraldehyde] in 0.1 M phosphate buffer (pH7.2) for 3 hr at room temperature. After washing with 0.1 M phosphate buffer, the embryos were postfixed with 1% osmium tetroxide in 0.1 M phosphate buffer for 1 hr at room temperature. The samples were washed with distilled water and dehydrated by a graded series of ethanol (50–100%) at 4°C. Finally, the embryos were embedded in durcupan resin and cut into ultrathin sections (70 nm thickness). The serial sections were collected on pioloform-coated single slot copper grids and

stained with uranyl acetate and lead citrate. They were then observed with a transmission electron microscope (JEM1010; JEOL Co., Japan). The serial section images were aligned with the IMOD software package (*Kremer et al., 1996*) as described previously (*Miyazaki et al., 2014*).

## Acknowledgements

We thank Dr. Makoto Suzuki for technical advises, Ms. S Yamada for technical assistance, Drs. G Goshima and CP Heisenberg for helpful discussion, and Dr. C Hudson for carefully reading our manuscript and Drs. A McDougall, T Momose, H Takahashi, D Jiang and Miho Suzuki for providing pRN3-PHGFP, pCX3-RfA-tdTomato, *Dsh* MO, *Ciona Dsh* construct and *Ciona* cDNAs, respectively. This work was supported by a grant from the ARC (1144) and the Agence Nationale de la Recherche (ANR-09-BLAN-0013-01) to HY and Grants-in-Aid for Scientific Research from the Japan Society for the Promotion of Science (13J03013, 22127007) to UN. The National Bio-Resource Project of the Ministry of Education, Culture, Sports, Science and Technology in Japan provided adult *Ciona intestinalis*. TN was supported by a postdoc fellowship from the Agence Nationale de la Recherche and Grant-in-Aid for JSPS Fellows. This study was supported by the collaborative research program of National Institute for Physiological Sciences, and Spectrography/Bioimaging Facility, NIBB Core Research Facilities.

## Additional information

### Funding

| Funder | Grant reference number | Author |
|---|---|---|
| Japan Society for the Promotion of Science | 13J03013 | Takefumi Negishi |
| Agence Nationale de la Recherche | ANR-09-BLAN-0013-01 | Hitoyoshi Yasuo |
| Fondation ARC pour la Recherche sur le Cancer | 1144 | Hitoyoshi Yasuo |
| Japan Society for the Promotion of Science | 22127007 | Naoto Ueno |

The funders had no role in study design, data collection and interpretation, or the decision to submit the work for publication.

### Author contributions

TN, Conception and design, Acquisition of data, Analysis and interpretation of data, Drafting or revising the article, Contributed unpublished essential data or reagents; NM, Acquisition of data, Analysis and interpretation of data, Drafting or revising the article, Contributed unpublished essential data or reagents; KM, Conception and design, Acquisition of data, Analysis and interpretation of data, Drafting or revising the article; HY, Conception and design, Analysis and interpretation of data, Drafting or revising the article, Contributed unpublished essential data or reagents; NU, Conception and design, Analysis and interpretation of data, Drafting or revising the article

### Author ORCIDs

Takefumi Negishi, http://orcid.org/0000-0002-0306-9578
Kazuyoshi Murata, http://orcid.org/0000-0001-9446-3652

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
