## [Decision Letter]

Thank you for submitting your article "Physical association between a novel plasma-membrane structure and centrosome orients cell division" for consideration by *eLife*. Your article has been favorably evaluated by K VijayRaghavan as the Senior editor and three reviewers, one of whom, Yukiko M Yamashita (Reviewer #1), is a member of our Board of Reviewing Editors.

The reviewers have discussed the reviews with one another and the Reviewing Editor has drafted this decision to help you prepare a revised submission.

Summary:

In this report, Negishi et al. describe a novel cell invagination during the early development of ascidian embryo that captures the centrosomes to orient the spindle. They describe this novel structure that is formed during oriented divisions of ascidian embryos (11th cell cycle). Long range centrosome migrations are known to be a feature of many large invertebrate embryonic cells, most notably ascidian embryos, but the origins of the directed force that drive this migration have been a mystery for over 100 years. Wilson for example talks about this type of behavior in his famous book from the last century. The apparent "action at a distance" of some cortical cue on the centrosome has always struck me as a very interesting problem. Here the authors suggest a novel solution whereby the membrane forms deep invaginations that reach out and attach to the centrosome, after which they exert elastic forces to pull the centrosome to the appropriate direction.

The morphological observations in this paper are truly striking and have been carried out with great thoroughness. The authors use a wide range of techniques from simple membrane dyes, to GFP tagged membrane proteins, to serial block face SEM, all of which reveal a highly novel structure composed of nested membranes from two adjacent cells diving deep into the interior of a cell to contact its centrosome. This invagination is associated with microtubules, which are apparently required for its formation or at least maintenance. This is thus a very interesting structure that many cell biologists, especially those interested in cytoskeleton-membrane interactions, will find very interesting. The images are of extremely high quality and all of the observations are rigorously quantified.

The reviewers agreed that this is a very interesting story describing a novel cellular structure that likely plays a critical role in spindle orientation during early development of ascidian embryos. The major concern was the lack of direct proof of a causal role of this invagination in spindle orientation. However, there is tremendous circumstantial evidence that this structure may play a direct mechanical role in centrosome migration. Given the novelty of the discovery, we believe that this paper will spark further research into the possible role of elastic membrane tethers in moving centrosomes, not only in ascidians but other systems as well. Given the complete mystery surrounding the long-range action at a distance underlying centrosome migration in invertebrate embryos, the demonstration of at least a potential mechanism is extremely welcome to the field. We therefore support the publication of this paper in *eLife*. But the critical lack of direct evidence must be appropriately acknowledged in the text.

Essential revisions:

Please go through the text to make it clear that the causal relationship (i.e. invagination orients spindle) is not proven. (Particularly because the laser ablation of invagination did not work in a way to provide insights).

The authors utilize disruption of planar cell polarity to test the idea of invagination being required for spindle orientation. Indeed, they observed that disruption of PCP causes invagination with randomized orientation, which nicely correlated with perturbed spindle orientation. But this does not necessarily prove that the invaginations play a causal role in moving the centrosomes, because it leaves open the possibility that there is some other down-stream process regulated by the planar cell polarity pathway that regulates centrosome movement independently of the invaginations. Thus, please make it clear that this is still correlational.

In addition, there have been a few suggestions that can potentially add mechanistic insights to the paper. However, after discussion, we felt that these are likely beyond the scope of the present manuscript. Therefore, the following suggestions are not required for acceptance of the manuscript. If the authors have such data already, they can be added. Or such possibilities may be discussed.

1) The actomyosin cortex appears to restrict the formation of those plasma membrane structures. It would be interesting to determine whether the cortex is weaker at the place where this structure forms, and how such potential anisotropy of the actomyosin cortex is established in the first place.

2) Microtubules appear to be required for the formation of those plasma membrane structures, but it remains unclear how microtubules generate the observed tensile forces. Presumably microtubule motors are involved, and it would thus be very interesting to identify any of such motors and their possible function in this process.

3) The orientation of the observed plasma membrane structure seems to determine the position of the primary cilium and the mitotic spindle, thus overriding other potential mechanisms, such as cell shape. It would be quite interesting to see whether e.g. cell division orientation is truly randomized in cases where the plasma membrane structure is cut, or whether other factors take over and bias division orientation.

---

## [Author Response]

Essential revisions:

Please go through the text to make it clear that the causal relationship (i.e. invagination orients spindle) is not proven. (Particularly because the laser ablation of invagination did not work in a way to provide insights).

We agree with this remark and we have modified the text accordingly. In particular,at the beginning of the Discussion section, we clearly state “[…] we could not directly demonstrate that the membrane invagination is required for mitotic spindle orientation […]”. Please note that we did not change the title of the manuscript, “Physical association between a novel plasma-membrane structure and centrosome orients cell division”. However, we are open to change it if necessary.

The authors utilize disruption of planar cell polarity to test the idea of invagination being required for spindle orientation. Indeed, they observed that disruption of PCP causes invagination with randomized orientation, which nicely correlated with perturbed spindle orientation. But this does not necessarily prove that the invaginations play a causal role in moving the centrosomes, because it leaves open the possibility that there is some other down-stream process regulated by the planar cell polarity pathway that regulates centrosome movement independently of the invaginations. Thus, please make it clear that this is still correlational.

We agree with this remark. Indeed, our Dsh-knockdown experiments cannot exclude the possibility that the PCP pathway regulates the centrosome dynamics and the directionality of invagination independently. In the last paragraph of the Discussion section, we now state “However, in this experiment, we cannot rule out the possibility that the PCP pathway affects centrosome behavior independently of this membrane structure”.

*In addition, there have been a few suggestions that can potentially add mechanistic insights to the paper. However, after discussion, we felt that these are likely beyond the scope of the present manuscript. Therefore, the following suggestions are not required for acceptance of the manuscript. If the authors have such data already, they can be added. Or such possibilities may be discussed.*

*1) The actomyosin cortex appears to restrict the formation of those plasma membrane structures. It would be interesting to determine whether the cortex is weaker at the place where this structure forms, and how such potential anisotropy of the actomyosin cortex is established in the first place.*

Yes, this is exactly what we are thinking. We observed that the number of invaginations was increased in Dsh-knockdown cells as well as in cytochalasin-treated cells, and it implies a possibility that the PCP pathway regulates anisotropy of the cortical actin. We now discuss this possibility in the last paragraph of the Discussion section:

“It will also be intriguing to investigate whether the PCP pathway regulates the asymmetric distribution of the actomyosin network and/or plus-end motors in ascidian epidermal cells.”

In the future, we plan to measure local differences and temporal changes of membrane stiffness using atomic force microscopy (AFM).

*2) Microtubules appear to be required for the formation of those plasma membrane structures, but it remains unclear how microtubules generate the observed tensile forces. Presumably microtubule motors are involved, and it would thus be very interesting to identify any of such motors and their possible function in this process.*

We agree with this comment; the physical association between membrane structure and microtubule is likely to be mediated by a motor protein. Indeed, the formation of the similar membrane invaginations in *C. elegans* zygotes depends on dynein (Redemann et al. 2010), it is possible that this plus-end motor protein is also involved in the process in ascidian epidermal cells. We now discuss this possibility in the last paragraph of the Discussion section: “The microtubule-dependent invagination process targeting the centrosome implies the involvement of plus-end-directed motor. Indeed, the generation of membrane invaginations in *C. elegans* one-cell embryos involves cortical dynein motors (Redemann et al., 2010)”. We are currently investigating the potential involvement of dynein in the ascidian membrane invagination but we do not yet have any solid data on this subject.

*3) The orientation of the observed plasma membrane structure seems to determine the position of the primary cilium and the mitotic spindle, thus overriding other potential mechanisms, such as cell shape. It would be quite interesting to see whether e.g. cell division orientation is truly randomized in cases where the plasma membrane structure is cut, or whether other factors take over and bias division orientation.*

As pointed out, during the 11th cell division, the ascidian epidermal cells divide along anterior-posterior axis overriding cell shape constraints and regardless of the direction of the 10th cell division axis. Our data support that this is likely to due to the polarized membrane invaginations controlling centrosome dynamics. Unfortunately, as described in the text, the immediate regeneration of membrane invagination after laser ablation did not allow us to test this possibility. Conversely, it would be very interesting if “experimentally-induced” membrane invaginations could affect the direction of cell division on other systems such as 10th cell cycle epidermal cells.